# Stochastic gene expression in auxin signaling in the floral meristem of *Arabidopsis thaliana*

Shuyao Kong [1,2,5], Byron Rusnak [1,2], Mingyuan Zhu[3,4] & Adrienne H. K. Roeder [1,2] ✉

Cells display striking stochasticity in gene expression, which plays an important role in development, disease, and regeneration. Previous studies have found stochastic gene expression in bacteria, yeast, and from constitutive promoters in Arabidopsis. However, most promoters are non-constitutive. Stochastic gene expression from non-constitutive promoters in a multicellular organism, especially those with key developmental roles, remains largely uncharacterized. Here, we report stochastic expression of auxin responsive genes in the Arabidopsis floral meristem, using promoter reporters of DR5, *ARABIDOPSIS HISTIDINE PHOSPHOTRANSFER PROTEIN6* (*AHP6*), and *DNA BINDING WITH ONE FINGER5.8* (*DOF5.8*). We find highly variable DR5 expression patterns among younger meristems. Such variability is largely attributed to stochastic expression of DR5, mainly influenced by cell-intrinsic molecular noise. Expression of *AHP6* and *DOF5.8* is also noisy, although their noise is lower and has distinct spatiotemporal patterns unlike DR5. Finally, we propose spatial averaging as a mechanism that buffers cellular gene expression noise, allowing the formation of robust global expression patterns. Our study reveals stochastic gene expression downstream of auxin signaling, a key developmental player. Thus, stochastic gene expression from non-constitutive promoters, including those involved in hormone signaling, is an ordinary part of multicellular life.

Genetically identical cells display high levels of stochasticity (random fluctuations) in gene expression, both spatially and temporally[1–6], which have been implicated in initiating cell fate specification in both animals and plants. For example, stochastic expression of key transcription factors such as Cdx2, Nanog, and Gata6 mediates stochastic determination of the trophectoderm, epiblast, and primitive endoderm fates, respectively, in the early mouse embryo[7–9]. These cells then sort to the correct positions[7–10]. In plants, stochastic expression of a transcription factor ARABIDOPSIS THALIANA MERISTEM LAYER 1 (ATML1) mediates

the differentiation of giant cells in the outer epidermis of sepals, which promotes flower opening[5,11,12]. In addition to cell fate determination, stochastic molecular noise is also utilized for decision making at the whole-organism level such as flowering[13,14] and seed germination[15,16], and during plant regeneration from protoplasts[17]. Besides development, in cancer, gene expression differences among tumor cells generate phenotypic variation, which allows clonal selection and cancer evolution[18,19]. Despite such multifaceted roles, studies of stochastic gene expression within the context of a multicellular organism are still scarce.

[1]Weill Institute for Cell and Molecular Biology, Cornell University, Ithaca, NY, USA. [2]Section of Plant Biology, School of Integrative Plant Science, Cornell University, Ithaca, NY, USA. [3]Department of Biology, Duke University, Durham, NC, USA. [4]Howard Hughes Medical Institute, Duke University, Durham, NC, USA. [5]Present address: Department of Pathology, Brigham and Women's Hospital; Department of Genetics, Harvard Medical School, Boston, MA, USA. ✉e-mail: ahr75@cornell.edu

Stochastic gene expression has been attributed to sources extrinsic and intrinsic to the cell[1,20]. Extrinsic noise includes fluctuating levels of external signals that different cells each receive, and heterogeneity in global cellular properties such as cell size, cell cycle stage, and the abundance of transcriptional and translational machinery. Intrinsic noise, on the other hand, refers to stochasticity in molecular processes such as chromatin state modifications, transcription, and translation. Extrinsic vs. intrinsic noise has been studied in various organisms using a dual reporter system, where two copies of the same promoter drive two reporters of different colors[1,2,4]. Across genetically identical cells, positively correlated variation of the two reporters, e.g., both highly expressed in one cell and low in another, indicates influence by extrinsic noise. On the other hand, independent variation, e.g. high expression of reporter 1 and low expression of reporter 2 in the same cell, suggests influence by intrinsic noise. However, so far, these studies have been limited to unicellular organisms, or gene expression from constitutive promoters such as UBIQUITIN (UBQ) 10 and 35S in Arabidopsis. Gene expression from non-constitutive promoters, such as those involved in stress adaptation, biotic interactions, and hormone signaling, is a crucial part of multicellular life. Stochasticity in their expression may impact the spatiotemporal dynamics of signaling pathways and potentially alter the phenotypic outcome. However, this subject remains largely understudied.

One of the plant hormones crucial to development is auxin[21–24]. Auxin is directionally transported by the PIN proteins to form auxin maxima which mediate organogenesis[21,25,26]. In the canonical auxin signaling pathway, Auxin binds to TRANSPORT INHIBITOR RESPONSE1/AUXIN SIGNALING F-BOX (TIR1/AFB) which in turn mediates the polyubiquitination and degradation of AUXIN/INDOLE-3-ACETIC ACID (Aux/IAA) proteins, releasing AUXIN RESPONSE FACTORS (ARFs) to trigger downstream transcriptional response[27]. To observe auxin response *in planta*, the DR5rev (referred to as DR5 below) reporter has been widely used[28]. DR5 is an artificial promoter consisting of nine auxin responsive elements in reverse orientation (GAGACA) fused to a minimal 35S promoter[29], and it reflects auxin-responsive gene expression mediated by ARFs. In addition to DR5, upstream auxin perception by TIR1/AFB can be revealed by the R2D2 reporter (*RPS5A::mDII-ntdTomato RPS5A::DII-n3×VENUS*)[30]. DII-n3×VENUS contains the DII motif from Aux/IAA that mediates auxin-dependent degradation[31–34], and mDII-ntdTomato contains a mutated non-degradable DII, so the mDII/DII signal ratio reflects auxin-dependent degradation of Aux/IAA by TIR1/AFB[35]. While these reporters have been widely used to reveal the level and pattern of auxin signaling, the extent of their stochasticity has not been characterized. Auxin signaling mediates primordium initiation, and robust primordium initiation requires precise spatiotemporal patterns of auxin signaling. Thus, stochasticity in auxin signaling patterns could affect the timing and/or pattern of primordium initiation[26,36]. On the other hand, stochastic expression of key developmental regulators could prime organogenesis[17]. These possibilities raise the question: To what extent is auxin signaling deterministic or stochastic?

Here, we characterize stochastic gene expression in DR5, a widely used auxin signaling reporter, in floral meristems of *Arabidopsis thaliana*. We also characterize stochastic gene expression from two endogenous auxin responsive promoters, *ARABIDOPSIS HISTIDINE PHOSPHOTRANSFER PROTEIN6* (*AHP6*) and *DNA BINDING WITH ONE FINGERS.8* (*DOF5.8*). We find that spatial patterns of DR5 expression are highly variable among younger meristems, which canalize into robust patterns in older meristems. In individual cells, however, DR5 expression is always stochastic, strongly influenced by intrinsic noise, and has no spatiotemporal patterns in noise amplitude. Expression of *AHP6* and *DOF5.8* is also stochastic, although their noise is weaker than DR5 and has spatiotemporal patterns. Finally, we propose that cellular noise in gene expression can be spatially averaged at the tissue scale in organs with sufficiently large numbers of cells to ensure robust global patterns.

## Results

### Auxin signaling gradually canalizes after exhibiting variability in young floral meristems

To characterize the stochasticity in auxin signaling, we imaged the Arabidopsis floral meristem, which allows us to characterize auxin pattern formation de novo. The floral meristem arises as a bulge on the inflorescence meristem periphery (stage 1). Soon it becomes separated from the inflorescence meristem (stage 2) and later initiates sepal primordia (stage 3)[37]. To better describe dynamical changes in patterns of auxin signaling during floral meristem development, we subdivided stage 1 and 2 based on meristem morphology and auxin signaling pattern revealed by DR5 (Fig. 1a–c). Stage 1a is when the meristem has just emerged from the inflorescence meristem and is flat on the upper (apical) surface. Stage 1b is when the upper surface of the meristem becomes convex, attaining positive Gaussian curvature (Fig. 1b). A boundary separating the meristem from the inflorescence meristem starts to form, as cells near the incipient boundary start to deform and their nuclei are stretched in the direction of the incipient boundary (Fig. 1a, inset). Stage 2a begins when the floral meristem becomes separated from the inflorescence meristem by a well-defined boundary. Viewed from the top, the meristem is wider in the lateral direction than the abaxial-adaxial (abaxial = away from the inflorescence meristem; adaxial = closer to the inflorescence meristem) direction. Auxin signaling typically forms two maxima in the incipient lateral sepal regions at this stage (Fig. 1c, d). Stage 2b is when the meristem expands in the abaxial-adaxial direction and becomes equally wide and tall. Auxin signaling typically forms two additional maxima in the incipient outer and inner sepal regions (Fig. 1c, d). Stage 2c is when the meristem further expands in the abaxial-adaxial direction and becomes taller than it is wide. Auxin signaling typically occurs in the incipient primordia of the inner whorls, in addition to the incipient sepals (Fig. 1c, d). Gaussian curvature in the incipient sepal-meristem boundaries starts to decrease (Fig. 1b, asterisks), in preparation for the emergence of sepal primordia in stage 3.

In early-stage meristems (stage 1a, 1b, and 2a), we found that auxin signaling is highly stochastic (Fig. 1c, arrowheads)[38]. Specifically, random patches of cells have strong auxin signaling compared to neighboring cells, and the location, size, and shape of these patches vary from bud to bud. Such stochasticity is not observed in older meristems (stage 2b and 2c), when auxin signaling becomes robustly concentrated in the four incipient sepal primordia, in addition to primordia of the inner whorls in stage 2c (Fig. 1c). Live imaging of DR5 revealed that the initial stochasticity in auxin signaling patterns in stage 1 gradually dampens, succeeded by four robustly positioned signaling maxima in stage 2b (Fig. 1e). These results suggest that spatial pattern of auxin signaling is highly variable in young floral meristems, which gradually canalizes to four robustly positioned maxima by the end of stage 2, prior to sepal initiation at these robust positions at stage 3[26,39].

To quantitatively characterize this finding, we aimed to quantify the average DR5 pattern and the variability of such pattern in each of these five developmental stages (Fig. 1f). We first measured DR5 signal in each cell of the upper epidermis (L1) of each bud. We used a ubiquitously expressed epidermal nuclear marker (*pATML1::H2B-TFP*) to detect and segment the nuclei of all the upper epidermal cells in which we quantified DR5 signal intensity. Buds were registered in a coordinate system, where the origin is set at the bud center, x-axis points laterally, and y-axis points abaxially. The upper epidermis was divided into equal-sized bins. Quantified nuclei were grouped into the bins according to their $(x, y)$ coordinates, and average signal intensity across all nuclei in each bin was calculated to generate a DR5 pattern heatmap for a given bud. The coordinate assignment, binning, and

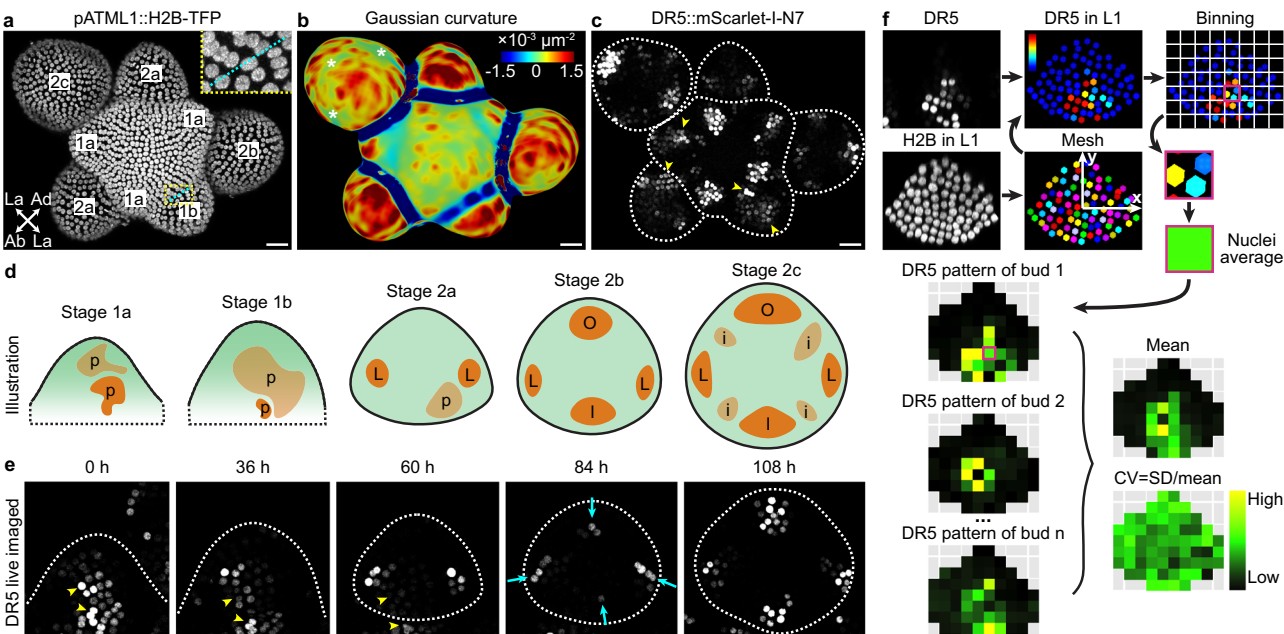

**Fig. 1 | Variable auxin signaling patterns in stage 1 floral meristems canalize into robustly positioned auxin maxima by late stage 2. a–c** Stage definitions. **a** Expression of *pATML1::H2B-TFP* in the epidermal (L1) nuclei showing tissue morphology. Stages are labeled on each meristem. Abaxial (Ab), adaxial (Ad), and lateral (La) directions of a stage 2a bud are labeled with arrows. Inset shows an incipient boundary (cyan) in stage 1b with deformed nuclei. **b** Gaussian curvature of extracted surface. Asterisks show decreasing Gaussian curvature in the incipient boundaries. **c** Auxin signaling patterns revealed by *DR5::mScarlet-I-N7*. Arrowheads show variable auxin patterns in stages 1a, 1b, and 2a. **d** Illustration of auxin signaling dynamics revealed by DR5. p, random patches of cells with high auxin signaling. L, incipient lateral sepals. O, incipient outer sepal. I, incipient inner sepal. i, incipient primordia of the inner whorls. **e** Live imaging of *DR5::mScarlet-I-N7* shows that variably localized patches of auxin signaling in stage 1 (arrowheads) dissipate, replaced by four robustly positioned auxin maxima in stage 2b and 2c. *n* = 4 buds. Scale bars in (**a–c**, **e**), 20 μm. **f** Method for quantifying mean and variability in reporter patterns used in Fig. 2. A mesh was made using *pATML1::H2B-TFP* signal in nuclei of the upper epidermis, within which DR5 was quantified. The upper epidermis was divided into equal-sized bins. Average signal intensity across nuclei in each bin was calculated to create a pattern heatmap for a given bud. Pattern heatmaps of multiple buds at the same stage were pooled to calculate the mean and coefficient of variation (CV) heatmaps.

calculation of pattern heatmap allows DR5 patterns in different buds to be quantitatively compared. DR5 pattern heatmaps from all buds of a given stage were pooled to generate a mean heatmap showing the average pattern, and a variability heatmap showing deviation from that average pattern. Variability is calculated as coefficient of variation (CV), which is standard deviation divided by mean. These summary heatmaps allow comparison of the mean and variability of DR5 patterns across different developmental stages (Fig. 2).

Using this method, we quantified the mean and variability of DR5 (Fig. 2a, c, d), and as a reference, *pATML1::H2B-TFP* (Fig. 2b–d), in the upper epidermis of WT floral meristems. We found that the DR5 pattern is highly variable among stage 1 meristems, forming randomly localized auxin signaling patches that are different from bud to bud (Fig. 2a). This variability gradually decreases toward stage 2, when all meristems form four auxin maxima robustly localized in the incipient sepals (Fig. 2a, c–e). Analysis of an independent insertion line of DR5 supports the same conclusion (Supplementary Fig. 1). In contrast, expression of *pATML1::H2B-TFP* is relatively uniform across the upper epidermis and has little variability between buds (Fig. 2b–e). Overall, these quantification results support the idea that auxin signaling revealed by DR5 is highly variable among stage 1 meristems and becomes robustly patterned in stage 2 prior to sepal initiation in stage 3.

**Limited contribution of auxin transport, level, and perception to the variability in auxin signaling**

We next asked where within the auxin signal transduction pathway the variability originates: in polar auxin transport, auxin level, auxin perception by TIR1/AFB – Aux/IAA, or the transcription of auxin responsive genes and reporters? To test the contribution of heterogeneity in

polar auxin transport and auxin level, we flooded DR5 reporter meristems with an excessive amount (100 μM) of 2,4-D, an auxin analog insensitive to polar auxin transport. Under such treatment, meristems do not uniformly upregulate auxin signaling across the bud periphery, but rather show randomly positioned patches of high auxin signaling (Supplementary Fig. 2a, arrowheads). Moreover, the same position in the meristem shows variable amount of auxin signaling across different replicates (Supplementary Fig. 2a, asterisks). When quantified, CV of global pattern in 2,4-D-treated meristems is uniform across stages (Supplementary Fig. 2b–d). Compared to mock-treated samples, 2,4-D treatment results in a 19.7–37.8% reduction in CV during stages 1a, 1b, and 2a, but leads to a 22.5% increase in stage 2c meristems. These results suggest that variability in polar auxin transport and auxin level may contribute partially to the observed variability of DR5 expression but does not fully account for it.

To test whether variability of auxin perception by TIR1/AFB – AUX/IAA explains the variability of DR5 expression, we observed R2D2, a reporter in which mDII/DII signal ratio reflects auxin perception[30]. We imaged R2D2 and calculated the mDII/DII ratio in each upper epidermal nucleus of the floral meristem from stage 1 to 2 (Fig. 3a). R2D2 shows a stereotypical pattern, where auxin perception concentrates first in the cryptic bract (a suppressed inflorescence leaf) (stage 1a), then in the adaxial-central region (stage 1b) which separates into two incipient lateral sepal primordia (stage 2a), then in the incipient outer and inner sepal primordia (stage 2b), and finally in the incipient primordia of the inner whorls (stage 2c) (Fig. 3b–d). The R2D2 pattern is robust and reproducible across buds of similar stage (Fig. 3b, e). In contrast to DR5 which shows random patches of high auxin signaling, especially in the adaxial-central region of stage 1 buds (Fig. 2a), R2D2 shows no such patches; particularly, the adaxial-

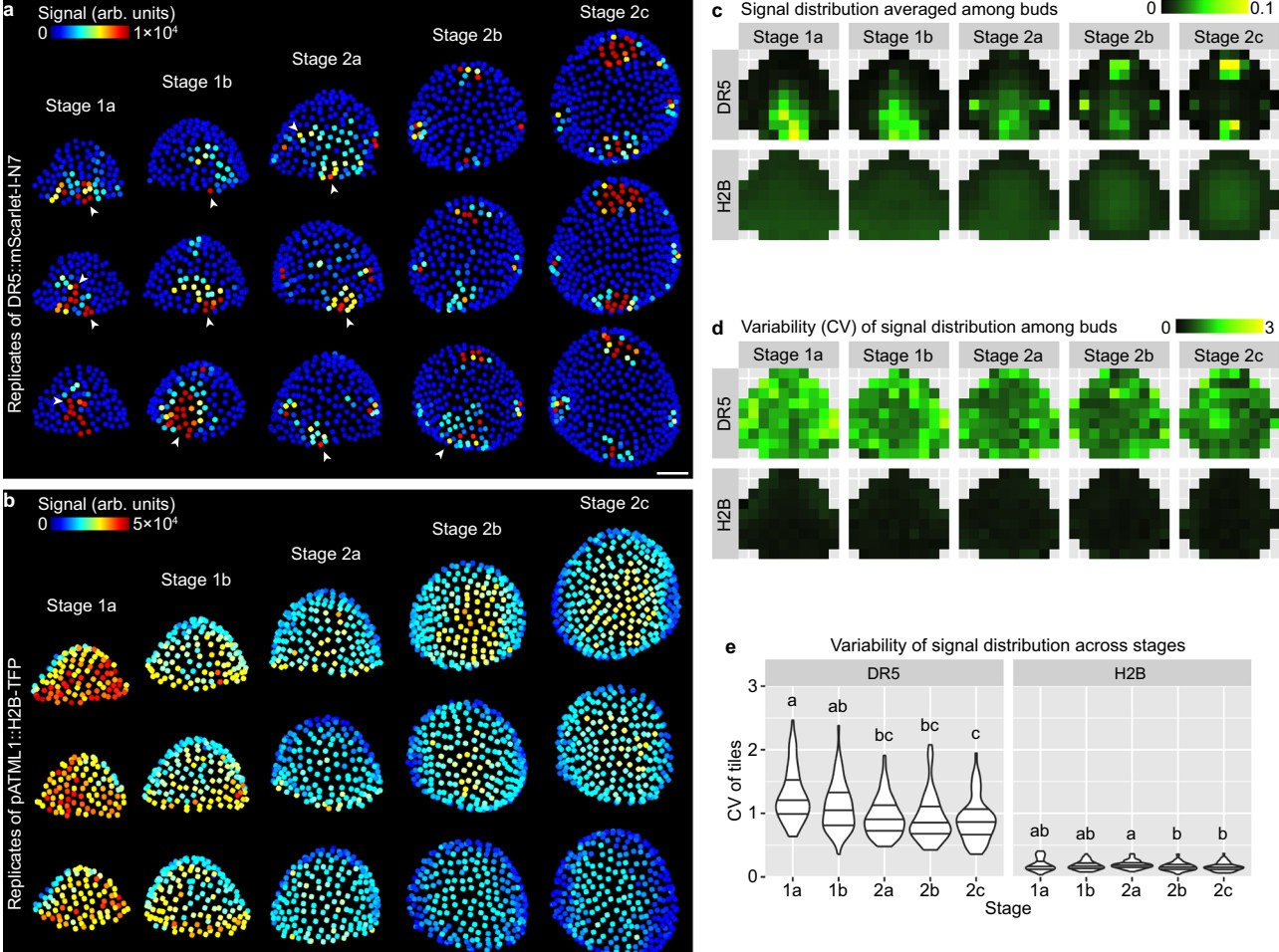

**Fig. 2 | Variability in global auxin signaling pattern among buds decreases with developmental stage. a** Three representative buds of each stage showing quantification of *DR5::mScarlet-I-N7* signal in the upper epidermis. Note highly variable patterns with sporadic patches of high-auxin-signaling cells in stage 1a, 1b, and 2a (arrowheads) which occur less frequently in stage 2b and 2c. **b** Three representative buds of each stage showing *pATML1::H2B-TFP* signal in the upper epidermis. Note patterns are relatively uniform and robust. Scale bars in (**a**, **b**), 20 μm. Mean (**c**) and variability (CV) (**d**, **e**) of signal distribution patterns calculated from all buds imaged: stage 1a, *n* = 22; stage 1b, *n* = 14; stage 2a, *n* = 17; stage 2b, *n* = 10; stage 2c,

*n* = 8. In (**e**), each data point is a CV value at an (x,y) tile of a given stage. Lines show quartiles. Letters denote statistical significance in pairwise two-sided permutation tests with Bonferroni's *p*-value adjustments. Adjusted *p*-values for DR5: 1a-1b, 0.061, 1a-2a, 0.000, 1a-2b, 0.000, 1a-2c, 0.000, 1b-2a, 0.051, 1b-2b, 0.059, 1b-2c, 0.006, 2a-2b, 1.000, 2a-2c, 1.000, 2b-2c, 1.000. Adjusted *p*-values for H2B: 1a-1b, 1.000, 1a-2a, 1.000, 1a-2b, 0.149, 1a-2c, 0.139, 1b-2a, 1.000, 1b-2b, 1.000, 1b-2c, 0.954, 2a-2b, 0.032, 2a-2c, 0.022, 2b-2c, 1.000. Note that CV of the DR5 pattern is much higher than *pATML1::H2B-TFP* and decreases with developmental stage. Also see Supplementary Fig. 1. Source data are provided as a Source Data file.

central region of stage 1a buds shows uniformly low mDII/DII ratio (Fig. 3b). Variability of the mDII/DII distribution pattern across buds is less than half of that of DR5 (Fig. 3e, f), similar to the extent of decrease in variability when stage 1a, 1b, and 2a meristems were treated with exogenous 2,4-D (Supplementary Fig. 2d). Overall, these results suggest that the variability in auxin signaling revealed by the DR5 reporter does not mainly come from the stochasticity in polar auxin transport, auxin level, or auxin perception; instead, it may mainly arise from stochastic molecular noise during the expression of auxin-responsive genes.

### Stochasticity in DR5 mainly arises from intrinsic molecular noise during gene expression

To test the idea that the observed auxin noise arises in the expression of auxin-responsive genes, we devised a dual-DR5 reporter system similar to other dual reporter systems previously published for characterization of stochastic gene expression (Fig. 4a)[1,2,4]. Each cell expresses two fluorophores VENUS and mScarlet-I, each driven by a DR5 promoter and nucleus-localized. Positively correlated variation of the two reporters across cells reveals stochastic gene

expression driven by extrinsic noise $\eta_{ext}$, while uncorrelated variation reveals contribution by intrinsic noise $\eta_{int}$. We imaged and quantified VENUS and mScarlet-I signal in all upper epidermal cells with the aid of an epidermal nuclear mesh generated from the *pATML1::H2B-TFP* reporter (Fig. 4b). To see whether the amplitude of cellular noise reduces throughout development, potentially contributing to the reduction of global pattern variability from stage 1a to 2c, we separately analyzed buds of each stage. We also note that positional cues, such as central zone (stem cell niche) vs. peripheral zone (competent zone for primordium initiation), incipient sepal vs. non-sepal regions, may define gene expression patterns in the floral meristem. We argue that such positional information, which is largely deterministic given tissue morphology, can also create correlated variation in the expression of the dual reporters but should not be included as part of extrinsic noise. We therefore employed the same coordinate assignment and binning methods as described above. Epidermal cells in the same (*x*, *y*) region of a bud were grouped in bins, and such cells in all buds of the same stage (*s*) were analyzed together for extrinsic and intrinsic noise, resulting in $\eta_{ext}|(x,y,s)$ and $\eta_{int}|(x,y,s)$ (Fig. 4b).

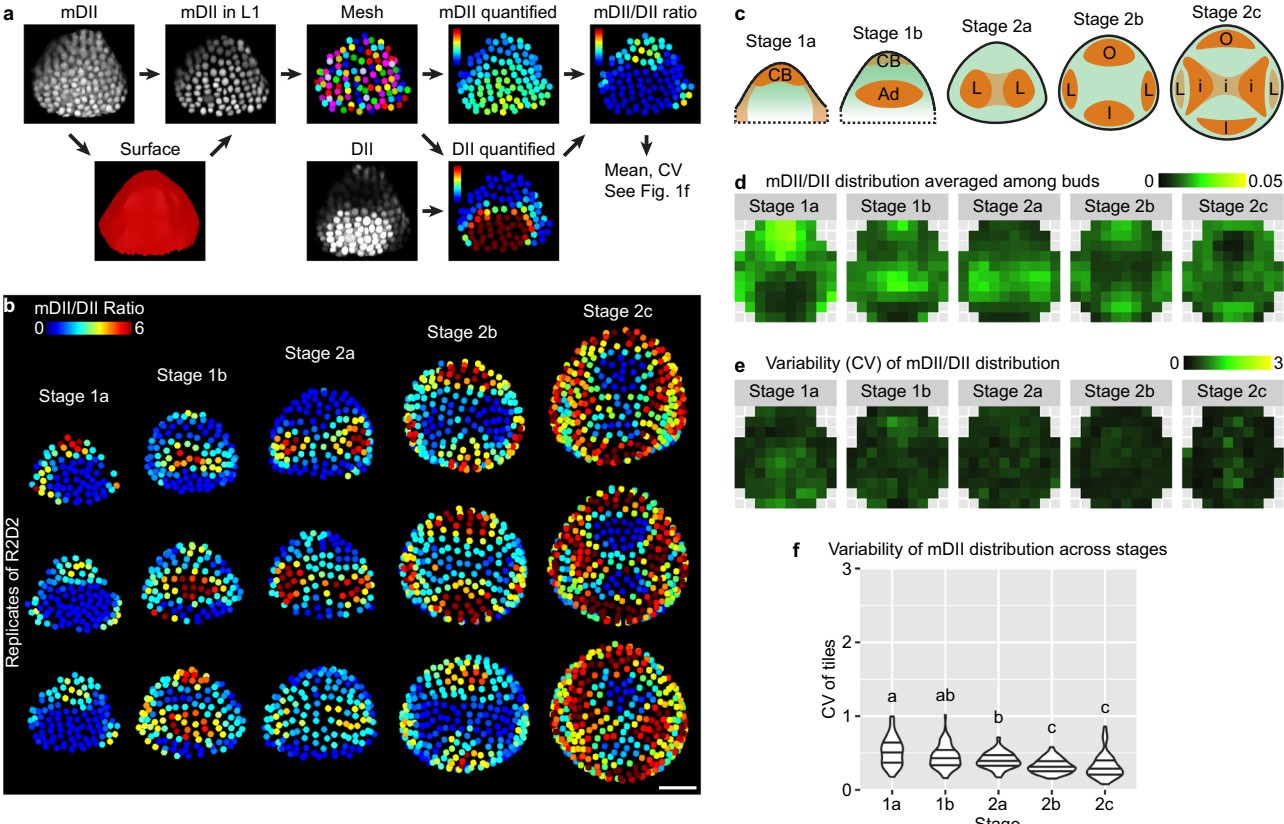

**Fig. 3 | Pattern of auxin perception is mostly robust. a** Method for quantifying spatial patterns of auxin perception. mDII signal was used to make a surface mesh, with which mDII signal in the upper epidermis was extracted, and an epidermal nuclear mesh was created. The mesh was used to quantify mDII and DII signal in each nucleus of the upper epidermis. mDII/DII ratio was calculated to reveal auxin perception in each cell. Mean and CV of the spatial pattern of mDII/DII ratio were calculated similar to Fig. 1f, to reflect the mean and variability of auxin perception patterns across buds. **b** mDII/DII ratio in three representative buds of each stage. Note the patterns are robust across different buds. Scale bar, 20 μm. **c** Illustration of representative patterns. CB cryptic bract, Ad adaxial region, L incipient lateral sepals, O incipient outer sepal, I incipient inner sepal, i incipient primordia of the inner whorls. Mean (**d**) and variability (CV) (**e**, **f**) of mDII/DII patterns calculated from all buds imaged: stage 1a, $n = 16$; stage 1b, $n = 10$; stage 2a, $n = 13$; stage 2b, $n = 12$; stage 2c, $n = 6$. In (**f**), each data point is a CV value at an (x,y) tile of a given stage. Lines show quartiles. Letters represent statistical significance in pairwise two-sided permutation tests with Bonferroni's $p$-value adjustments. Adjusted $p$-values: 1a-1b, 0.053, 1a-2a, 0.000, 1a-2b, 0.000, 1a-2c, 0.000, 1b-2a, 0.222, 1b-2b, 0.000, 1b-2c, 0.000, 2a-2b, 0.000, 2a-2c, 0.003, 2b-2c, 1.000. Note that the variability in mDII/DII ratio is much lower than DR5 and also decreases with developmental stage. Also see Supplementary Fig. 2. Source data are provided as a Source Data file.

Consistent with our observation that DR5 is highly stochastic in stage 1a, 1b, and 2a (Fig. 2a), we found highly stochastic expression of both DR5 reporters in the dual-reporter system, forming random patches (Fig. 4c). Importantly, these patches are largely uncorrelated between the two reporters, suggesting heavy influence by intrinsic molecular noise in gene expression (Fig. 4c). In stage 2b and 2c where DR5 becomes concentrated in four signaling maxima robustly positioned at the four incipient sepals, we surprisingly also found highly uncorrelated variation in the expression of the dual reporters. Within each signaling maxima, although both reporters are expressed in the region, not all cells uniformly express both reporters at the same level. Cells expressing only one of the reporters but not the other are prevalent (Fig. 4c). When extrinsic and intrinsic noise are calculated across all upper epidermal cells of all buds irrespective of cellular position and bud developmental stage, they are similar ($\eta_{ext} = 1.822$; $\eta_{int} = 1.689$; Fig. 4d), both much higher than noise caused by instrument and measurement errors measured by imaging identical fluorescent beads ($\eta_{ext} = 0.0335$; $\eta_{int} = 0.0397$; Supplementary Fig. 3). However, the level and pattern of auxin signaling are different in incipient sepal vs. non-sepal regions and between stages (Fig. 4e); these deterministic differences should be excluded during the calculation of extrinsic noise. We thus calculated noise conditioned on bin position and bud stage, $\eta_{ext}|(x, y, s)$ and $\eta_{int}|(x, y, s)$. Under this calculation, we found much higher intrinsic noise than extrinsic noise

(Fig. 4f, g). Neither of them has a distinguishable spatial pattern, nor do they strongly correlate with total signal intensity (Fig. 4h, i). In agreement with our observation of uncorrelated variation in the dual reporters even at stage 2c (Fig. 4c), cellular noise does not decrease with developmental stage, but stays unchanged or even slightly increases (Fig. 4j, k). This is in contrast with the observation that the global DR5 pattern becomes more robust from stage 1 to 2 (Fig. 2e). These conclusions were supported by an independent transgenic line of *DR5::mScarlet-I-N7* together with *DR5::3×VENUS-N7* (Supplementary Fig. 4). Overall, these results suggest that auxin signaling revealed by DR5 is heavily impacted by cell-intrinsic molecular noise in gene expression, and the noise persists from stage 1 till the end of stage 2.

**Stochastic noise influences the expression of AHP6 and DOF5.8 to a lower extent and in a position-dependent manner**

We next wondered how widespread this stochasticity is in the expression of auxin-responsive genes. To this end, we created dual reporters of *AHP6* and *DOF5.8*, two auxin-responsive genes downstream of *MONOPTEROS/ARF5*[36,40,41]. *AHP6* is expressed in lateral organ primordia and acts non-cell autonomously to regulate phyllotaxy in the shoot apical meristem[36,42]. *DOF5.8* is highly expressed in the vascular precursor cells and is important for vascular differentiation[43,44]. We observed dual reporters of *AHP6* and *DOF5.8* in floral meristems to see whether their expression is influenced by stochastic noise.

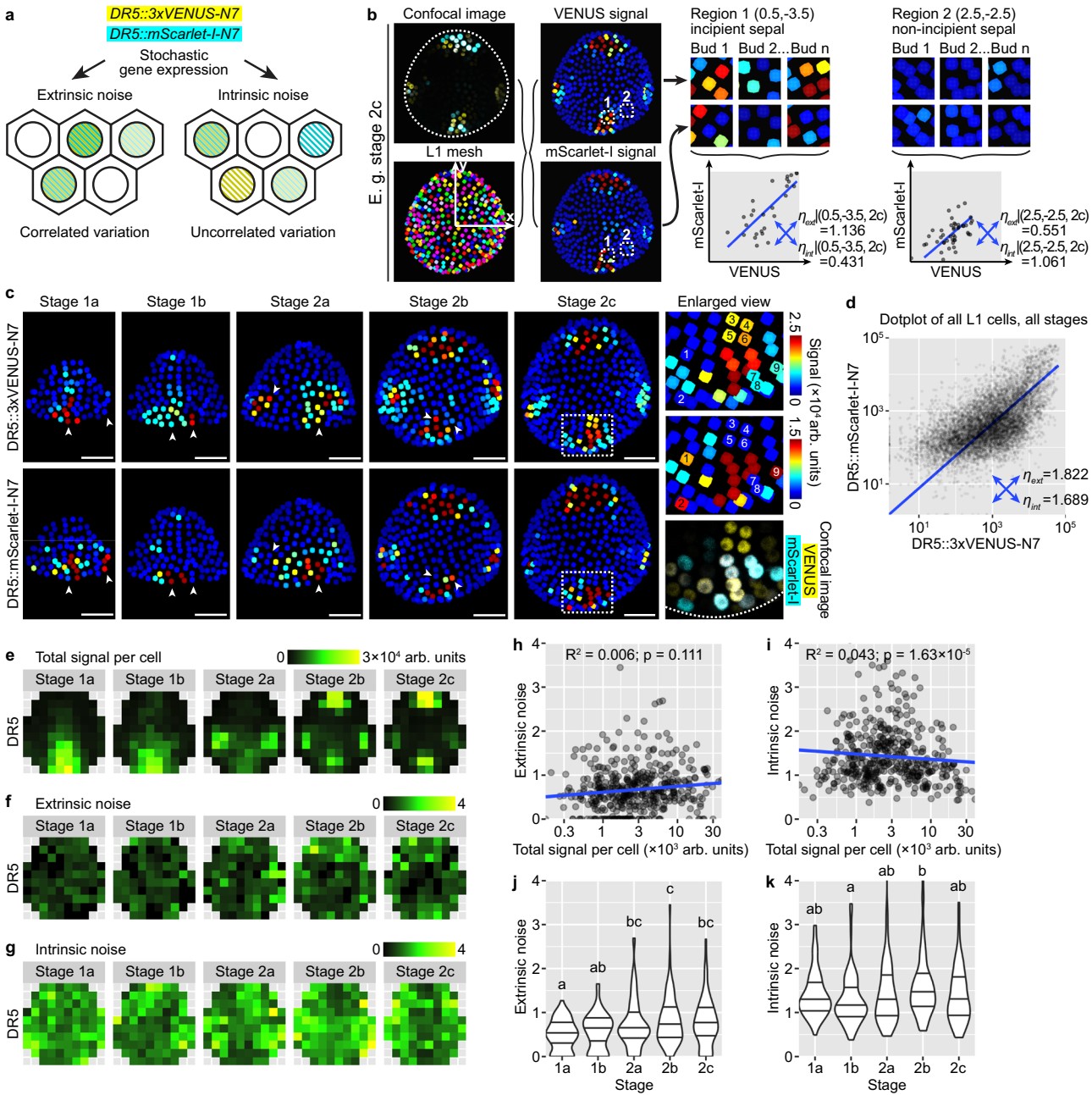

**Fig. 4 | Cell-intrinsic noise affects expression from the auxin-responsive promoter DR5. a** A dual-DR5 reporter system reveals cell-extrinsic and cell-intrinsic contributions to stochastic gene expression. **b** Method for noise calculation from dual-reporter image stacks. **c** Dual-reporter signal from representative buds. Arrowheads show cells strongly expressing one reporter but not the other, indicating influence by intrinsic noise. In enlarged view of stage 2c, 1–9 show nine nuclei that express only one reporter. Note that in stage 2b and 2c, although four auxin maxima form at robust positions, cells in these auxin maxima are still strongly influenced by stochastic gene expression. Scale bars, 20 μm. **d** Dot plot of dual reporter signals of all cells from buds of all stages. Number of buds: stage 1a, *n* = 17; stage 1b, *n* = 10; stage 2a, *n* = 9; stage 2b, *n* = 9; stage 2c, *n* = 7. **e** Summed signal from both channels, averaged across all cells in each tile. Extrinsic (**f**) and intrinsic (**g**) noise calculated from all cells in each tile, which lack apparent spatial patterns.

Relation of extrinsic and intrinsic noise to total signal per cell (**h, i**) and developmental stage (**j, k**). Each data point is a noise value at an (x,y) tile of a given stage. In (**j, k**), lines show quartiles; letters denote statistical significance in pairwise two-sided permutation tests with Bonferroni's *p*-value adjustments. Adjusted *p*-values for (**j**): 1a-1b, 1.000, 1a-2a, 0.006, 1a-2b, 0.000, 1a-2c, 0.001, 1b-2a, 0.173, 1b-2b, 0.009, 1b-2c, 0.068, 2a-2b, 1.000, 2a-2c, 1.000, 2b-2c, 1.000. Adjusted *p*-values for (**k**): 1a-1b, 1.000, 1a-2a, 1.000, 1a-2b, 0.643, 1a-2c, 1.000, 1b-2a, 1.000, 1b-2b, 0.023, 1b-2c, 1.000, 2a-2b, 1.000, 2a-2c, 1.000, 2b-2c, 0.511. In contrast to variability in global DR5 pattern across buds which decreases from stage 1 to 2, cellular noise either barely changes (intrinsic noise) or slightly increases (extrinsic noise) with developmental stage. Also see Supplementary Figs. 3, 4. Source data are provided as a Source Data file.

Expression of *AHP6* in the epidermis of the floral meristem follows a slightly different pattern than DR5 and R2D2. It first appears in the abaxial-central region of the bud (stage 1), which later disappears, replaced by high expression in the incipient lateral sepals (stage 2a) and finally all four incipient sepals (stage 2b and 2c) (Fig. 5a). In contrast to DR5, the two *AHP6* reporters show highly correlated variation among cells, across all developmental stages (Fig. 5a, b). Noise calculation conditioned on bin position and bud stage shows that expression of *AHP6* is influenced by similar levels of extrinsic and intrinsic noise, although both weaker than those of DR5 (Fig. 5c–e).

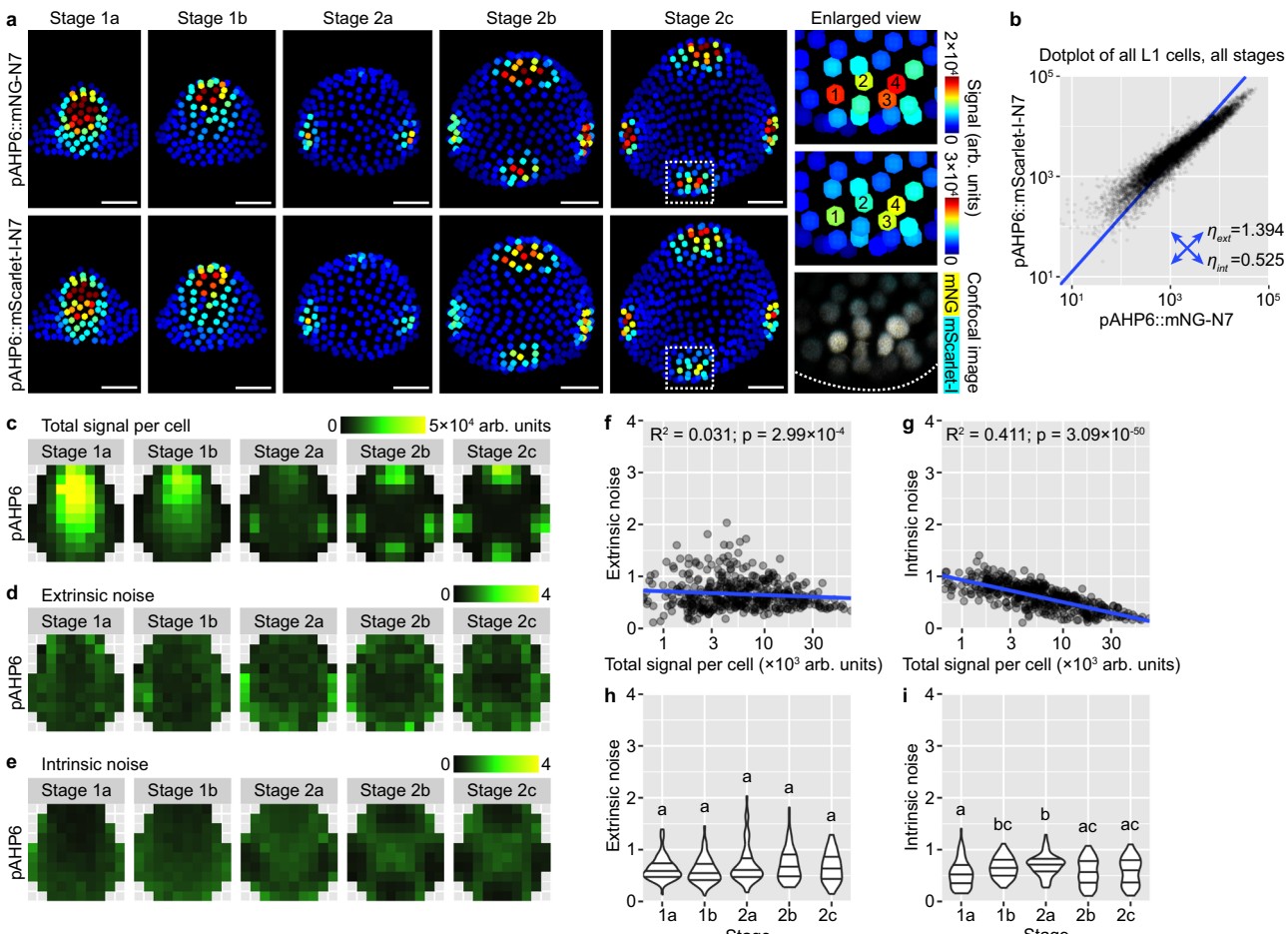

**Fig. 5 | Stochastic gene expression from the AHP6 promoter. a** Dual-reporter signal in representative buds. Note that signals from the two reporters largely overlap, indicating low intrinsic noise. In enlarged view of stage 2c, 1–4 label four nuclei with strong expression of both reporters. Scale bars, 20 μm. **b** Dot plot of dual reporter signals of all cells from buds of all stages. Number of buds: stage 1a, $n = 11$; stage 1b, $n = 11$; stage 2a, $n = 12$; stage 2b, $n = 8$; stage 2c, $n = 6$. **c** Summed signal from both channels, averaged across all cells in each tile. Extrinsic (**d**) and intrinsic (**e**) noise calculated from all cells in each tile. Note that extrinsic noise in *AHP6* expression is higher in the peripheral zone than the central zone; intrinsic noise is higher in the non-sepal regions than incipient sepal regions. **f**, **g** Extrinsic and intrinsic noise are negatively correlated with total signal per cell. **h**, **i** Relation of extrinsic and intrinsic noise to developmental stage. Note that extrinsic noise does not change through stages; intrinsic noise first increases and reaches the highest in stage 1-2 transition, and then decreases in stage 2b and 2c. In (**f**–**i**), each data point is a noise value at an (x,y) tile of a given stage. In (**h**, **i**), lines show quartiles; letters denote statistical significance in pairwise two-sided permutation tests with Bonferroni's *p*-value adjustments. Adjusted *p*-values for (**h**): 1a-1b, 1.000, 1a-2a, 1.000, 1a-2b, 1.000, 1a-2c, 1.000, 1b-2a, 0.089, 1b-2b, 0.089, 1b-2c, 0.815, 2a-2b, 1.000, 2a-2c, 1.000, 2b-2c, 1.000. Adjusted *p*-values for (**i**): 1a-1b, 0.013, 1a-2a, 0.000, 1a-2b, 1.000, 1a-2c, 1.000, 1b-2a, 1.000, 1b-2b, 0.160, 1b-2c, 0.593, 2a-2b, 0.001, 2a-2c, 0.014, 2b-2c, 1.000. Also see Supplementary Fig. 5. Source data are provided as a Source Data file.

Interestingly, both extrinsic and intrinsic noise in *AHP6* expression display relations with spatial location of the cell. Extrinsic noise is higher in the bud peripheral zone than in the central zone (Fig. 5d) and slightly negatively correlates with combined signal intensity (Fig. 5f). Intrinsic noise is strongly negatively correlated with combined signal intensity (Fig. 5g) and lower in regions with high *AHP6* expression (for example, the incipient sepal primordia; compare Fig. 5e with Fig. 5c). When buds of different stages were compared, extrinsic noise does not change with developmental stage (Fig. 5h), while intrinsic noise increases from stage 1a to 2a when *AHP6* expression in the abaxial-central region declines, and decreases from stage 2a to 2c when *AHP6* expression in the incipient sepals becomes established (compare Fig. 5i with Fig. 5c). These findings are supported by an independent dual-reporter line of *AHP6* (Supplementary Fig. 5). Thus, expression of *AHP6* in the floral meristem epidermis is noisy, although less so than DR5, and the amplitude of noise follows distinct spatiotemporal patterns not seen in DR5. These differences from DR5 may imply additional control of *AHP6* expression that reduces noise to improve developmental robustness.

Expression of *DOF5.8* in the floral meristem epidermis follows similar spatiotemporal patterns as *AHP6* (Fig. 6a). It is initially highly expressed in the central zone (stage 1), which gradually declines and is replaced by high expression in the incipient sepal primordia (stage 2) (Fig. 6a). Although this global pattern is relatively robust like AHP6, it is still influenced by intrinsic molecular noise like DR5. Cells that strongly express one of the reporters but not the other are prevalent in all stages (Fig. 6a). The amplitude of cellular noise when all cells are quantified together is in between that of DR5 and AHP6 (Fig. 6b). When cells are grouped by bin position and bud stage, extrinsic and intrinsic noise are similar in amplitude and, again, in between the noise amplitude of DR5 and AHP6 (Fig. 6c–e). When different positions in the bud are compared, extrinsic noise is higher in the peripheral zone than the central zone, similar to AHP6 (Fig. 6d), while intrinsic noise does not have a clear spatial pattern (Fig. 6e). The amplitude of noise is not strongly affected by total signal intensity (Fig. 6f, g) or bud developmental stage (Fig. 6h, i), except that extrinsic noise slightly decreases from stage 1a to 2a and then slightly increases in stage 2b and 2c. These conclusions are

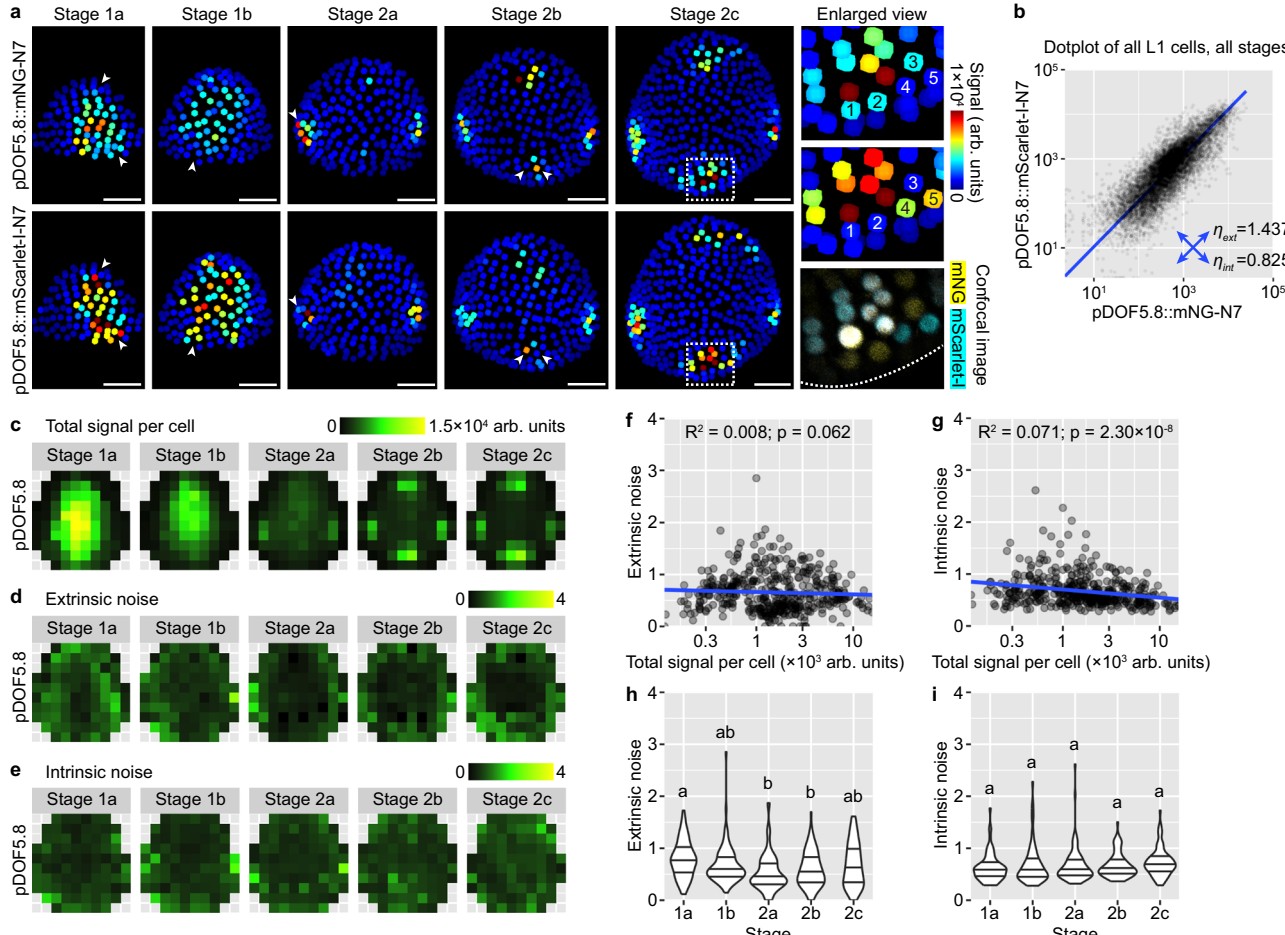

**Fig. 6 | Stochastic gene expression from the DOF5.8 promoter. a** Dual-reporter signal in representative buds. Note cells expressing one of the reporters but not the other (arrowheads), indicating influence by intrinsic noise. In enlarged view of stage 2c, 1–5 label five nuclei that only express one of the reporters. Similar to DR5, although auxin maxima form at robust positions in stage 2b and 2c, cells in these auxin maxima are still strongly influenced by stochastic gene expression. Scale bars, 20 μm. **b** Dot plot of dual-reporter signals of all cells from buds of all stages. Number of buds: stage 1a, *n* = 16; stage 1b, *n* = 13; stage 2a, *n* = 11; stage 2b, *n* = 8; stage 2c, *n* = 7. **c** Summed signal from both channels, averaged across all cells in each tile. Extrinsic (**d**) and intrinsic (**e**) noise calculated from all cells in each tile. Note that extrinsic noise in *DOF5.8* expression is higher in the peripheral zone than

the central zone, similar to *AHP6*; intrinsic noise has no apparent spatial pattern. Relation of extrinsic and intrinsic noise to total signal per cell (**f**, **g**) and developmental stage (**h**, **i**). Each data point is a noise value at an (x,y) tile of a given stage. In (**h**, **i**), lines show quartiles; letters denote statistical significance in pairwise two-sided permutation tests with Bonferroni's *p*-value adjustments. Adjusted *p*-values for (**h**): 1a-1b, 0.481, 1a-2a, 0.000, 1a-2b, 0.002, 1a-2c, 0.561, 1b-2a, 0.058, 1b-2b, 0.968, 1b-2c, 1.000, 2a-2b, 1.000, 2a-2c, 0.116, 2b-2c, 1.000. Adjusted *p*-values for (**i**): 1a-2c, 0.095, others, 1.000. Note that extrinsic noise is highest in stage 1a and decreases from stage 1b; intrinsic noise is not stage dependent. Also see Supplementary Figs. 6, 7. Source data are provided as a Source Data file.

largely supported by an independent insertion line of *DOF5.8* dual reporters, except that intrinsic noise is higher in that line, reaching a level comparable to DR5, particularly in the incipient sepal regions (Supplementary Fig. 6). Thus, extrinsic and intrinsic sources of noise influence *DOF5.8* expression similar to DR5, but like *AHP6*, the noise amplitude varies with spatial location and bud developmental stage.

DOF5.8 is important for vasculature differentiation[43,44]. Thus, in addition to the epidermis, we characterized stochastic gene expression in L2 and L3 cells expressing *DOF5.8*, which are likely vasculature precursor cells (Supplementary Fig. 7a). We imaged and quantified mNG and mScarlet signals in all L1, L2, and L3 cells in which expression of either reporter was detected (Supplementary Fig. 7b). We only calculated intrinsic noise because the omission of cells that do not express either reporter would cause an underestimation of extrinsic noise. For this analysis, we divided stage 1 and 2 meristems into three stages based on how many files of cells express *DOF5.8*. 1-file stage roughly corresponds to stage 1. 2-file stage roughly corresponds to the end of stage 1b and the beginning

of stage 2a. 4-file stage roughly corresponds to the rest of stage 2 (Supplementary Fig. 7a). We found that intrinsic noise in the expression of *DOF5.8* in vasculature precursor cells changes with bud developmental stage and cellular position. Buds at 2-file stage have the highest intrinsic noise of all three stages (Supplementary Fig. 7c). At 4-file stage, the cell files at the incipient outer and inner sepals have higher intrinsic noise than the cell files at the incipient lateral sepals (Supplementary Fig. 7d). When cells from different tissue layers were separately analyzed (Supplementary Fig. 7e), we found that cells in deeper tissue layers have less intrinsic noise than cells closer to the bud surface (Supplementary Fig. 7f). In summary, expression of two auxin-responsive genes, *AHP6* and *DOF5.8*, is influenced by sources of noise extrinsic and intrinsic to the cell; unlike DR5, the amplitude of their noise is lower and varies with cellular position and bud developmental stage, which implies noise-reducing control mechanisms. How gene expression noise in endogenous auxin-responsive genes is controlled, and how development is still robust despite the remaining noise, are intriguing open questions.

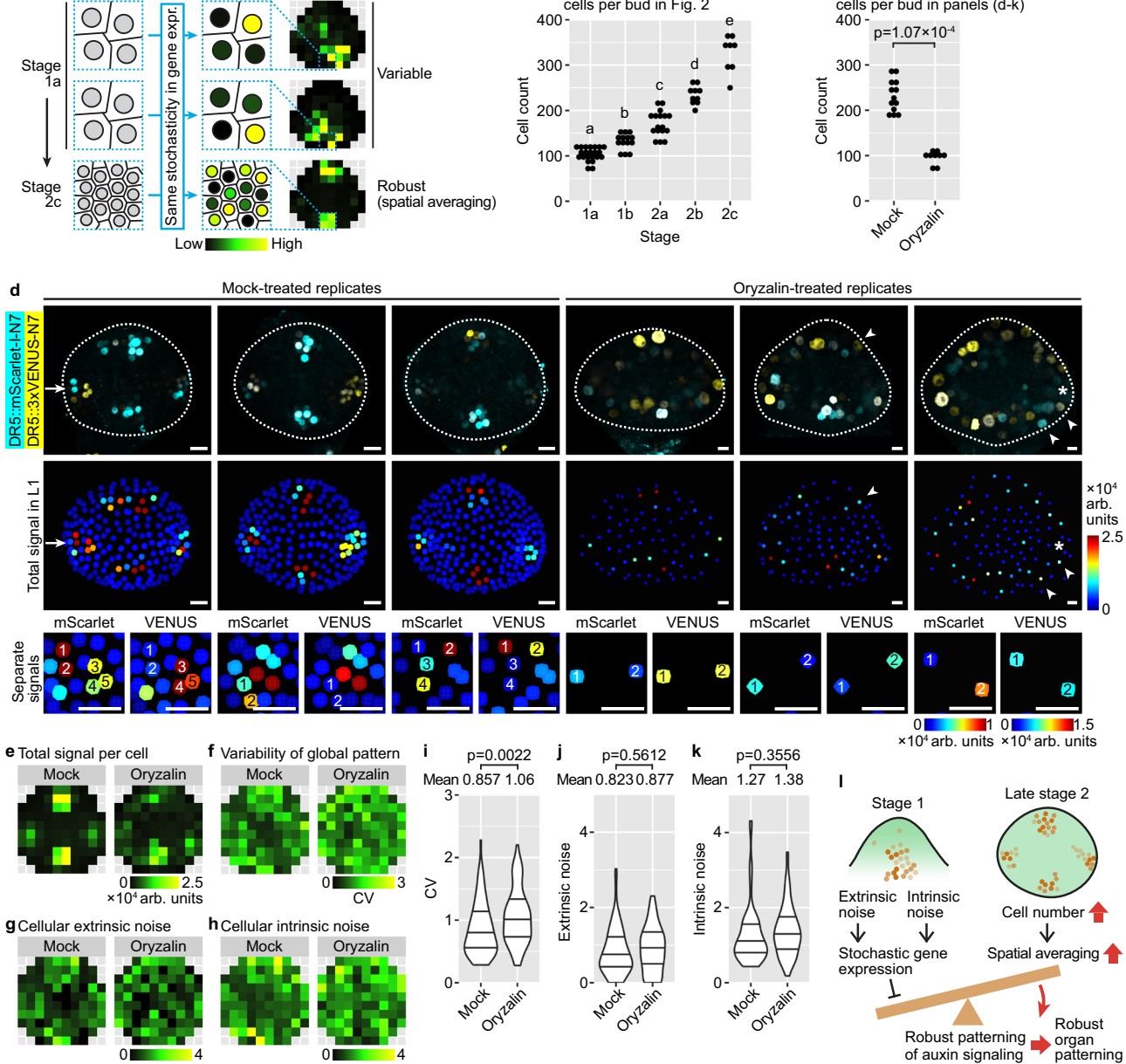

**Fig. 7 | Spatial averaging of stochastic gene expression creates tissue-wide pattern robustness. a** Spatial averaging. As the bud grows, a larger cell number allows stochastic gene expression at the cellular level to be averaged out among neighboring cells, creating robust tissue-wide patterns. **b** Number of upper epidermal cells increases from stage 1a to 2c. 1a, $n = 22$; 1b, $n = 14$; 2a, $n = 17$; 2b, $n = 10$; 2c, $n = 8$. Letters denote statistical significance from Tukey's multiple comparison. $p$-values: 1a-1b, $5.75 \times 10^{-3}$; 1b-2a, $1.69 \times 10^{-4}$; others, $<1 \times 10^{-4}$. **c** Oryzalin treatment decreases cell number in the upper epidermis. $p$-value, two-sided Wilcoxon's test. **d** Three representative buds of DR5 dual-reporter, mock or oryzalin-treated. Top, overlaid channels. Middle, summed signal. Arrow, an incipient sepal primordium in which not all cells express both copies of DR5 but still makes an auxin signaling maximum. Asterisk, an incipient sepal primordium that has no cells expressing DR5 and does not make an auxin signaling maximum. Arrowheads, cells outside incipient sepal primordia expressing DR5 stochastically, making exogenous auxin

signaling maxima. Bottom, separate channels showing intrinsic noise. Scale bars, 10 μm. **e** Summed signal. **f** Variability of global DR5 pattern across buds. **g** Cellular extrinsic noise. **h** Cellular intrinsic noise. **i** Variability of global DR5 pattern increases upon oryzalin treatment. Cellular extrinsic noise (**j**) and intrinsic noise (**k**) are not changed by oryzalin. For (**c**–**k**), to ensure match of developmental stage, only the oldest stage 2 bud from each inflorescence was analyzed. Mock, $n = 13$; oryzalin, $n = 9$. For (**i**–**k**), each data point represents an (x,y) tile. Lines show quartiles. $p$-values, two-sided permutation tests. **l** Conceptual model. Extrinsic and intrinsic sources of noise contribute to stochastic gene expression, which can hamper the formation of robust global patterns. As the bud grows bigger toward the end of stage 2, more cells allow spatial averaging of stochastic gene expression among neighboring cells, which permits the formation of robust global patterns of auxin signaling prior to robust initiation of sepals at stage 3. Also see Supplementary Fig. 8. Source data are provided as a Source Data file.

## Spatial averaging of stochastic gene expression creates tissue-wide pattern robustness

The auxin signaling pattern, as revealed by DR5, is highly variable in the epidermis of floral meristems from stage 1a to 2a, but becomes concentrated in four robustly positioned incipient sepal primordia in stage 2b and 2c (Fig. 2a). However, DR5 expression is strongly influenced by

stochastic noise throughout stages 1 and 2 (Fig. 4). How does the global pattern of auxin signaling become robust despite persistent molecular noise? We hypothesized an effect of cell number on the robustness of global patterns against stochastic gene expression (Fig. 7a). Specifically, younger meristems have fewer cells (Fig. 7b), and it is easy for stochastic gene expression at the cellular level to create variability in

the global pattern among buds. As the meristem grows bigger and more cells are available for sculpting the global pattern (Fig. 7b), stochastic gene expression in each cell still persists, but it is averaged out when cell neighborhoods are considered, so it less affects the global pattern. For example, in an incipient sepal primordia of a stage 2c bud (Fig. 4c), despite stochastic gene expression on a cell-by-cell basis, because there are enough cells in the incipient primordia region, there will always be cells that express auxin-responsive genes, mediating the output of auxin signaling and organ initiation in that region (Fig. 7a). On the other hand, in boundary regions between incipient sepals, despite cells that stochastically expresses auxin responsive genes, most other cells do not express them, keeping the overall auxin signaling level low. Thus, all buds of stage 2c will have auxin signaling maxima only in the four incipient sepal regions, ensuring robustness in global auxin signaling pattern.

To test this hypothesis, we decreased cell number in buds of late stage 2 by treating them with oryzalin, which depolymerizes microtubules and thus inhibits cell division (Fig. 7c). Because oryzalin may have an effect on cell expansion[45,46], to match the developmental stage of buds being compared, we selected the oldest stage 2 bud (with no sepal primordia) in each inflorescence. Mock-treated buds show four robustly localized auxin signaling maxima (Fig. 7d, top and middle rows), although cells are still strongly affected by stochastic gene expression (Fig. 7d, bottom row). In each incipient sepal primordium, due to stochasticity, not all cells express auxin-responsive genes; however, there are always some cells that express them, making an auxin signaling maximum (Fig. 7d, arrow). In contrast, in oryzalin-treated buds, due to a decrease in cell number, some incipient sepal primordia do not have any cells expressing DR5, thus not making an auxin signaling maximum (Fig. 7d, asterisk). Cells outside the four incipient sepal primordia regions can stochastically express DR5, making exogenous auxin signaling maxima that may be connected to other maxima in incipient sepal regions nearby (Fig. 7d, arrowheads). Thus, in oryzalin-treated buds, global pattern of auxin signaling is strongly influenced by stochastic gene expression, being both less precise (Fig. 7e) and more variable between different buds (Fig. 7f, i). This increase in global pattern variability is not accompanied by an increase in gene expression noise at the cellular level (Fig. 7g, h, j, k), suggesting that the decrease in cell number likely contributes to the increase in variability of global pattern among buds.

As an independent way to test the effects of decreasing cell numbers on the global pattern variability, we treated meristems with hydroxyurea, a cell cycle inhibitor. Such treatment results in a 33% decrease in the number of cells of the upper epidermis of stage 2 meristems (Supplementary Fig. 8a), which was statistically significant albeit less effective than oryzalin (a 59% decrease, Fig. 7c). Hydroxyurea-treated meristems show absence of auxin signaling in some regions that normally initiate sepal primordia (asterisk, Supplementary Fig. 8b), and sporadic patches of auxin signaling in regions outside the normal incipient sepal primordia (arrowheads, Supplementary Fig. 8b), similar to oryzalin-treated meristems. Hydroxyurea treatment slightly (11%) although non-significantly increases the variability of global pattern, while cellular noise remains unchanged (Supplementary Fig. 8c–i). Overall, these results support our hypothesis that stochastic gene expression, while still present, less affects the global pattern when cell number is sufficiently large.

## Discussion

Stochastic gene expression is a widespread phenomenon[1–6]. In Arabidopsis, stochastic gene expression has been found for the 35S and UBQ10 promoters[4]. It was not known whether such stochasticity also exists in response to signals such as hormones. In this study, we characterized stochastic gene expression from auxin-responsive promoters, DR5, *pAHP6*, and *pDOF5.8*, in floral meristems of Arabidopsis. We found high variability in DR5 pattern in young (stage 1 to 2a) floral

meristems, which canalizes to robust pattern in late stage 2 (Figs. 1 and 2). Such variability mainly comes from stochastic expression of auxin-responsive genes, largely influenced by intrinsic molecular noise (Fig. 4). Upstream processes – such as noise in polar auxin transport, heterogeneity in auxin level, and noise in auxin perception – make a minor contribution at most to the variability in DR5 pattern (Fig. 3, Supplementary Fig. 2). Expression of *AHP6* and *DOF5.8* is similarly stochastic, though to a lesser extent than DR5 and with distinct spatiotemporal patterns of noise (Figs. 5 and 6). Finally, we propose that the increase in cell number from stage 1 to 2 promotes spatial averaging of stochastic gene expression, producing robust global patterns of auxin signaling that underlie robust organ initiation in stage 3 (Fig. 7l). Our work revealed stochastic gene expression in response to auxin signaling, a process central to plant development, laying the foundation for future studies of how such stochasticity is buffered or even utilized during plant development[47,48].

Previous publications have used dual reporter systems to study the origins of stochastic gene expression[1,2,4]. Notably, in leaves and root tips of Arabidopsis plants, gene expression from ubiquitous promoters such as 35S and UBQ10 is influenced by extrinsic and intrinsic noise, $\eta_{ext}$ and $\eta_{int}$, and it was found that extrinsic noise has a much larger influence than intrinsic noise[4]. Here, we used a similar dual reporter system to study stochastic gene expression in DR5 in the epidermis of floral meristems. When all upper epidermal cells were considered, irrespective of location and bud developmental stage, extrinsic noise was indeed higher than intrinsic noise (Fig. 4d). However, unlike ubiquitous promoters, auxin-responsive promoters are influenced by auxin signal, which is strongly dependent on cell position within an organ and developmental stage (Fig. 3). Thus, we argue that deterministic information such as cellular position and developmental stage should be separated from more stochastic sources of heterogeneity such as the transcriptional and translational capability of a cell. Thus, for calculation of cellular noise in DR5 expression, we binned nuclei based on their position $(x, y)$ and bud stage $(s)$, to get $\eta_{ext}|(x, y, s)$ and $\eta_{int}|(x, y, s)$. Under this calculation, we found that the variability in DR5 expression is mostly influenced by intrinsic noise (Fig. 4f, g). Such binning also allowed us to conclude that cellular noise in DR5 expression does not have a spatial pattern (Fig. 4f, g) and is largely independent of developmental stage (Fig. 4j, k), a conclusion unclear to us when all cells were analyzed together (Fig. 4d). Thus, we established a pipeline to analyze the spatiotemporal patterns of stochastic gene expression from non-ubiquitous promoters such as hormone-responsive ones, where all cells cannot be considered as coming from the same population. We hope this pipeline will be useful for studying gene expression noise in a variety of processes beyond auxin signaling.

Besides DR5, we found that the expression of *AHP6* and *DOF5.8*, two auxin-responsive genes, is also influenced by stochastic noise. Compared to DR5, their noise is lower and has spatial patterns not seen in DR5. Extrinsic noise of *AHP6* and *DOF5.8* expression is higher in the peripheral zone than the central zone (Figs. 5d and 6d). Intrinsic noise of *AHP6* is lower in the incipient sepal primordia where *AHP6* expression concentrates (Fig. 5e). Intrinsic noise of *DOF5.8* is higher in the incipient outer and inner sepals than the incipient lateral sepals, and decreases with distance from the bud surface (Supplementary Fig. 7). These differences in the amplitude and spatial patterns of noise may have origins in promoter architecture. (Supplementary Fig. 9)[49] *pDR5* consists of nine tandem TGTCTC repeats in reverse orientation sandwiching eight pyrimidine-rich motifs (Y-patches)[50] proximal (-191 to -98 bp) to the transcription start site (TSS). Such close packing of AuxREs may confer high affinity to ARFs, facilitating their cooperative binding[51] and permitting sporadic transcription. On the other hand, *pAHP6* and *pDOF5.8* contain fewer AuxREs, which are more widely spaced and all outside the proximal promoter (−250 to +1bp). Such arrangement may set a higher threshold for ARF-mediated transcription and serve to

filter out stochastic noise. Moreover, *pAHP6* and *pDOF5.8* contains diverse AuxREs, including TGTCGG, TGTCCC, TGTCAC, and TGTSTSBC, in addition to TGTCTC which is the only AuxRE in *pDR5*. Such diversity may promote the heterodimerization between ARFs with different AuxRE preferences, thus filtering out noise from the stochastic fluctuation of the level of a particular ARF. Lastly, *pAHP6* and *pDOF5.8* have a wide range of coupling elements surrounding the AuxREs[50]. These coupling elements can serve as binding sites for ARF partners, which may exert control in noise level. Future studies are needed to better understand the effect of promoter architecture on the stochasticity of gene expression during multicellular development.

If DR5 is noisier than endogenous auxin-responsive promoters, can it still be used for studies of auxin signaling? We would argue yes, but with two caveats: (1) DR5 expression does not always correlate with expression of all auxin-responsive genes (for example, compare stage 1 in Figs. 4e, 5c and 6c), so activation of developmental programs downstream of auxin signaling needs to be checked using reporters of genes in these specific programs. (2) DR5 expression at the cellular level is highly noisy, but global DR5 pattern is robust due to spatial averaging among neighboring cells (Fig. 7). Thus, DR5 expression can be used as a proxy for auxin signaling only on a cell neighborhood-basis, not on an individual cell basis.

While the DR5 pattern is highly variable between different buds of stage 1, it becomes robust in late stage 2 (Fig. 2), in contrast to persistent extrinsic and intrinsic noise (Fig. 4). Based on our experimental result that decreasing cell number in stage 2 buds increases the variability of global DR5 patterning (Fig. 7c–k, Supplementary Fig. 8), we postulate that stochastic gene expression persists on a cell-by-cell basis but is averaged out at the tissue scale when cell number increases from stage 1 to 2, making global pattern robust and reproducible among buds (Fig. 7l). Our idea parallels previous studies of plant organ growth, where fast-growing cells are interspersed among slow-growing cells, so that growth rate at the tissue scale is constant, ensuring robust final organ size and shape[3,6,52]. The effect of reducing cell number on reducing the robustness of global gene expression pattern may explain the previous observations that organ initiation from the floral meristem becomes less robust when cell number was genetically reduced[53], or when organ initiation was premature[38]. Thus, organisms seem to strike a balance between producing enough cells for robust morphogenesis and the potential time and resource costs of producing cells.

# Methods

## Constructs

The following constructs were previously published: *DR5::3×VENUS-N7*[28], *pATML1::H2B-TFP*[54], and R2D2[30].

The following constructs were made by VectorBuilder using a pPBV binary vector backbone. *DR5::mScarlet-I-N7* was designed by putting together a *DR5rev* promoter, a plant Kozak sequence (AAAA; same below), an mScarlet-I coding sequence without stop codon, a linker (ATTGCTGCAGCGGCC; same below), an N7 coding sequence with stop codon, and an *OCS* terminator. The plant selection marker was Kanamycin (*pNOS::Kozak:KanR:NOSter*) (Supplementary Data 1).

*pAHP6::mNG-N7* was designed by putting together a *AHP6* promoter (1597 bp before the start codon), a plant Kozak sequence, an mNeonGreen coding sequence without stop codon, a linker, an N7 coding sequence with stop codon, and a 500 bp *AHP6* terminator. The plant selection marker was Basta (*pNOS::Kozak:BlpR:NOSter*) (Supplementary Data 2). *pAHP6::mScarlet-I-N7* was similarly designed, except that the fluorescent protein was mScarlet-I and the plant selection marker was Kanamycin (*pNOS::Kozak:KanR:NOSter*) (Supplementary Data 3).

*pDOF5.8::mNG-N7* was designed by putting together a *DOF5.8* promoter (1934 bp before the start codon), a plant Kozak sequence, an mNeonGreen coding sequence without stop codon, a linker, an N7

coding sequence with stop codon, and a 439 bp *DOF5.8* terminator. The plant selection marker was Basta (*pNOS::Kozak:BlpR:NOSter*) (Supplementary Data 4). *pDOF5.8::mScarlet-I-N7* was similarly designed, except that the fluorescent protein was mScarlet-I and the plant selection marker was Kanamycin (*pNOS::Kozak:KanR:NOSter*) (Supplementary Data 5).

## Plant material

All Arabidopsis (*Arabidopsis thaliana*, RRID:NCBITaxon_3702) plants were in Col-0 background (WT). For single reporters of *DR5::mScarlet-I-N7*, *pAHP6::mScarlet-I-N7*, and *pDOF5.8::mScarlet-I-N7*, plants already carrying the *pATML1::H2B-TFP* construct were transformed with the respective mScarlet-I constructs using the floral dip method, and T2 plants were imaged. For the DR5 dual reporter, plants carrying both *pATML1::H2B-TFP* and *DR5::mScarlet-I-N7* were crossed with *DR5::3×VENUS-N7* plants[28], and F1 plants were imaged. For AHP6 and DOF5.8 dual reporters, Col-0 plants were transformed with the respective mNG constructs. T1 plants were crossed with plants carrying both *pATML1::H2B-TFP* and the respective mScarlet markers, and F1 plants were imaged. For R2D2, the original line in Columbia-Utrecht background[30] was backcrossed with Col-0 twice before using.

All T1 transformants were screened using dual-PCR for single-copy insertion lines (Supplementary Fig. 10; see below for methods)[55]. These single-insertion lines were confirmed by counting segregation ratio of antibiotic resistance in T2 (Supplementary Table 1).

## Plant growth conditions

Seeds were sown in wetted Lamber Mix LM-111 soil and stratified at 4 °C for 2–7 days. Plants were grown under 16 h light – 8 h dark cycles (fluorescent light, ~100 μmol m⁻¹ s⁻¹) at 22 °C in a Percival walk-in growth chamber.

## Dual PCR

A 520 bp region of an endogenous single-copy gene (*HPPD*) was amplified as an internal reference. A pair of primers amplifying a ~600 bp region of the transgene (*BAR* or *mScarlet-I*) were designed so that they have similar melting temperatures as the primers for *HPPD*. Genomic DNA was extracted using the CTAB method from T1 plants to be tested. PCR reactions were assembled using ExTaq DNA polymerase (TaKaRa) according to the manufacturer's suggestions, except that four primers were used (two *HPPD* primers and two transgene primers), each at 500 nm concentration. The reactions were run at 58 °C annealing and 1 min extension, for 20 cycles to avoid saturation. The product was then electrophorized in an Ethidium Bromide-containing gel and visualized under UV light. The relative intensities of the band for *HPPD* and the band for the transgene were used to infer transgene copy number in each T1 plant. Samples in which the transgene band is brighter than the endogenous band are likely from multi-copy T1 plants. Samples in which the transgene band is slightly fainter than the endogenous band are likely from single-copy T1 plants. Primer sequences can be found in Supplementary Table 2.

## Flower staging

Flower buds of stage 1-2[37] were further divided into sub-stages as below (Fig. 1a–c):

Stage 1 is when the meristem has just emerged but not yet separated from the inflorescence meristem, which is further divided into two stages. In stage 1a, the meristem emerges as a bulge on the side of the inflorescence meristem. The apical (top) side of the new meristem is still flat. It is hard to tell where the boundary is between the new meristem and the inflorescence meristem. In stage 1b, the apical side of the new meristem bulges out and attains positive Gaussian curvature. A boundary starts to form between the new meristem and the inflorescence meristem, as cells at the boundary start to fold in and their nuclei start to deform along the incipient boundary. However, the new

meristem has not completely separated from the inflorescence meristem at this stage.

Stage 2 is when the new meristem has separated from the inflorescence meristem, but no sepal primordia have formed, which is further divided into three stages. In stage 2a, the new meristem has just become separated from the inflorescence meristem by a well-defined boundary. Viewed from the top, it is wider in the lateral direction than the abaxial-adaxial direction. Auxin signaling revealed by the DR5 reporter usually shows two auxin maxima in the incipient lateral sepals. In stage 2b, the bud has grown wider in the abaxial-adaxial direction and attains equal aspect ratio viewed from the top. Auxin signaling usually concentrates into four maxima, one in each incipient sepal. In stage 2c, the bud further expands to be wider in the abaxial-adaxial direction than in the lateral direction. Auxin signaling occurs in the incipient primordia of the inner whorls, in addition to the incipient sepals. Gaussian curvature of the bud surface starts to change in the incipient sepal boundaries; however, sepal primordia formation, which marks the end of stage 2 and onset of stage 3, has not yet occurred.

We note that this staging system is based on meristems growing in vivo. Prolonged (several days of) tissue culture in vitro may change meristem morphology and blur the boundaries between stages 2a, 2b, and 2c.

For analysis of *DOF5.8* expression in the vascular precursors (Supplementary Fig. 7), because the number of cell files expressing *DOF5.8* does not strictly correlate with the staging system described above, we divided buds based on the number of cell files expressing *DOF5.8* (1-file stage, 2-file stage, and 4-file stage).

### Drug treatments

Oryzalin (Sigma cat. no. 45-36182-100MG-EA) was dissolved in DMSO to make an 84 mM stock. The stock was added to autoclaved and cooled inflorescence culture medium to a final concentration of 50 μM. Dissected inflorescences were cultured on this medium for 4 days before imaging. For mock, inflorescences were cultured on medium containing 0.06% DMSO. For image analysis, only the oldest stage 2 bud (without sepals) in each inflorescence was analyzed; only cells in the upper epidermis were counted.

Hydroxyurea (ThermoFisher cat. no. A10831-03) was added directly to autoclaved and cooled inflorescence culture medium to a final concentration of 100 mM. Dissected inflorescences were cultured on this medium for 6 days before imaging, being transferred to freshly prepared medium every 2–3 days. For mock, inflorescences were cultured on medium containing no hydroxyurea. For image analysis, only the oldest stage 2 bud (without sepals) in each inflorescence was analyzed; only cells in the upper epidermis were counted.

2,4-Dichlorophenoxyacetic acid (PhytoTech Labs cat. no. D299) was dissolved in 100% ethanol to create a 100 mM stock solution. This stock was then added to autoclaved and cooled inflorescence culture medium to a final concentration of 100 μM. Dissected inflorescences were cultured on this medium for 2 days before imaging. For mock, inflorescences were cultured on medium containing 0.1% ethanol.

### Confocal microscopy

When inflorescences were 5–10 cm tall, they were cut and dissected with a Dumont tweezer (Electron Microscopy Sciences, style 5, cat. no. 72701-D) to remove buds older than stage 9 or 10. They were then inserted upright into a small petri dish (VWR, 60 ×15 mm) containing inflorescence culture medium (1/2 MS, 1% (w/v) sucrose, 1x Gamborg vitamin mixture, 0.1% (v/v) plant preservative mixture (Plant Cell Technology), 1% (w/v) agarose, pH 5.8) so that the base of the explant was in the medium and the top of the explant with all the buds was outside. Sterile water was added on top of the medium, surrounding the buds, to prevent loss of turgor (which makes further dissection difficult). These inflorescences were further dissected while inserted in the media to remove buds older than stage 3, leaving all buds desirable

for imaging (stage 1 and 2) uncovered. More water was then added on top of the media, and the explants were imaged under a Leica Stellaris 5 upright confocal microscope with an HC FLUORTAR L VISIR 25×/0.95 water-dipping lens and a resonance scanner (8000 Hz, bi-directional scanning, and 8-times line averaging). The following laser and wavelength settings were used. TFP, excitation 448 nm, emission 453–520 nm. VENUS and mNG, excitation 514 nm, emission 519–550 nm. mScarlet-I, excitation 561 nm, emission 566–650 nm. Frame switching was used to prevent interference between channels. For live imaging, between time points, samples were put in a growth chamber with 16 h light – 8 h dark cycles.

For fluorescent beads imaging to quantify noise caused by instrument and measurement errors (Supplementary Fig. 3), 4.0 μm beads in TetraSpeck™ Fluorescent Microspheres Sampler Kit (Invitrogen cat. no. T7284) was diluted 1:100 in water, pipetted onto a glass slide, air dried, and imaged using mNG and mScarlet-I settings same as above. Only single beads were used for analyses of extrinsic and intrinsic noise.

### Image processing

Image stacks were exported from the Leica software to ImageJ, deinterleaved (run("Deinterleave", "how=2") or run("Deinterleave", "how=3") depending on the number of channels), cropped to desired dimensions, and saved in tif format. The tif stacks were then imported into MorphoGraphX. Signal from outside the bud being analyzed, such as other buds, inflorescence meristem, etc., was removed using the Voxel Edit tool.

To create a surface from *pATML1::H2B-TFP*, expressed in the epidermis, the TFP channel stack was loaded into MorphoGraphX and following processes were run: Gaussian Blur Stack (x sigma = y sigma = z sigma = 3 μm), Edge Detect (threshold = 6000, multiplier = 2.0, adapt factor = 0.3, fill value = 30000), Marching Cubes Surface (cube size = 6 μm, threshold = 20,000), Subdivide, Smooth Mesh (passes = 5, walls only = no). Only the upper epidermis (above the equator) was used for analysis; nuclei from below the equator were removed using the Voxel Edit tool.

To create a Gaussian Curvature heatmap from the surface, the process Project Mesh Curvature was run (type = Gaussian, neighborhood = 10 μm, autoscale = no, min curv = −0.0015, max curv = 0.0015).

To create a mesh of nuclei from *pATML1::H2B-TFP*, the TFP channel stack was loaded into MorphoGraphX and the following processes were run: Gaussian Blur Stack (x sigma = y sigma = z sigma = 0.8 μm), Local Maxima (x radius = y radius = z radius = 0.5 μm, start label = 2, threshold = 3000, value = 60000), Mesh From Local Maxima (radius = 2 μm). Occasionally, the program detects more than one local maxima per nuclei; in that case, only one local maximum selected randomly was retained in each nucleus. The mesh was then positioned so that the center of the bud was at the origin, x-axis pointed laterally, y-axis pointed abaxially, and z-axis pointed apically. The polar coordinates of each analyzed nucleus were recorded using the process Polar Coord (central axis = z) and then converted into Cartesian coordinates.

To quantify fluorescent signal in the L1 using mesh created from *pATML1::H2B-TFP*, each channel (VENUS, mScarlet-I, or mNG) was loaded into MorphoGraphX together with the mesh. The following processes were run: Project signal (use absolute = no, min dist = 0 μm, max dist = 2 μm, min signal = 0.0, max signal = 100000.0, max project = no), Signal Total, Heat Map Set Range, Save to CSV Extended. Note that because only the upper epidermis was used to create the mesh, only signal in the upper epidermis was projected and quantified; signal from below the equator was automatically discarded.

For the *DOF5.8* dual reporters, to quantify fluorescent signal in all cells expressing at least one of the two reporters (mNG and mScarlet-I), image stacks from both channels were combined by running the process Combine Stacks (method = add). The following processes

were run to create a mesh of all nuclei expressing either reporter: Gaussian Blur Stack (x sigma = y sigma = z sigma = 0.8 μm), Local Maxima (x radius = y radius = z radius = 0.5 μm, start label = 2, threshold = 8000, value = 60,000), Mesh From Local Maxima (radius = 2 μm). Distance of each nucleus to the bud upper surface was calculated by loading the surface mesh in Mesh 2 and running the process Distance to Mesh. The bud was positioned, polar and cartesian coordinates of each nucleus were calculated, and reporter signals were quantified same as described above.

The resulting csv files, containing coordinates and signal intensity of each quantified nucleus, were then imported into R for downstream analyses. To account for differences in the size of each bud, the 80th percentile of the radial coordinate (distance to the Z axis) was used to normalize the x and y coordinates of each nucleus. Nuclei that were more than 1.3 times the 80th percentile of the radial coordinate were discarded. A binwidth of 0.25 was used to bin nuclei in the x and y directions, resulting in $0.25 \times 0.25$ tiles. By binning nuclei into tiles of the same coordinate system, signal distribution patterns of different buds can be directly compared. Subsequent analyses were all based on these tiles.

For calculation of mean and variability (CV) of global patterns of *DR5::mScarlet-I-N7* and *pATML1::H2B-TFP* reporters, within each bud, mean signal of all nuclei within each tile was calculated, normalized to the total signal of all tiles in that bud, to get signal distribution among the tiles ("pattern heatmap"). Such heatmaps were generated for all buds of a given stage (e.g., stage 2a). For each tile position, mean and CV (SD divided by mean) were calculated across all buds at that stage. This generates the mean and CV heatmaps, which summarize the average pattern and variability of pattern of a given reporter across all buds at a given stage. Tiles with data from 2 or fewer buds were discarded. For R2D2, ratio between mDII and DII signals in each nucleus was calculated to represent auxin perception, and such mDII/DII ratio was used to calculate the mean and CV heatmaps similar to described above.

For calculation of extrinsic and intrinsic noise in dual reporters of DR5, AHP6, and *DOF5.8*, we used previously published formulae[1,2,4]

$$\eta_{ext}^2 \equiv \frac{\langle c_1 c_2 \rangle - \langle c_1 \rangle \langle c_2 \rangle}{\langle c_1 \rangle \langle c_2 \rangle} \qquad (1)$$

$$\eta_{int}^2 \equiv \frac{\langle (c_1 - c_2)^2 \rangle}{2 \langle c_1 \rangle \langle c_2 \rangle} \qquad (2)$$

where $c_1$ and $c_2$ are signal intensities from channel 1 and channel 2, respectively, and angled rackets denote means for all quantified nuclei in the upper epidermis of all buds of all stages. Note that these definitions treat all cells as if they come from the same cell population, disregarding their spatial location and bud stage. These definitions summarize the contribution of extrinsic and intrinsic sources of noise to heterogeneity of gene expression among cells. However, such extrinsic noise includes deterministic sources of heterogeneity, such as incipient sepal region vs. non-sepal region, central zone vs. peripheral zone, and developmental stage of the bud, together with more stochastic sources of variability such as cell cycle stage and the transcriptional and translational capability of a cell. Moreover, these definitions cannot address whether there are differences in the amplitude of noise between different spatial locations in a bud and across developmental stages. To address these shortcomings, we calculated extrinsic and intrinsic noise for all cells at a given tile position in all buds at a given stage

$$\eta_{ext}^2|(x,y,s) \equiv \frac{\langle (c_1 c_2)|(x,y,s) \rangle - \langle c_1|(x,y,s) \rangle \langle c_2|(x,y,s) \rangle}{\langle c_1|(x,y,s) \rangle \langle c_2|(x,y,s) \rangle} \qquad (3)$$

$$\eta_{int}^2|(x,y,s) \equiv \frac{\langle (c_1 - c_2)^2|(x,y,s) \rangle}{2 \langle c_1|(x,y,s) \rangle \langle c_2|(x,y,s) \rangle} \qquad (4)$$

where $(x,y)$ is the tile position, and $s$ is the developmental stage of the bud (1a, 1b, 2a, 2b, or 2c). These noise values were then plotted with respect to tile position and stage.

## Scanning of AuxRE and coupling motifs in promoters

Promoters used in making the reporter constructs (443 bp for *pDR5*, 1597 bp for *pAHP6*, and 1934 bp for *pDOF5.8*) were scanned for the AuxREs TGTCNC, TGTCGG, and TGTSTSBC, as well as the coupling motifs Y-patch, AuxRE-like, and ABRE-like[50]. FIMO version 5.5.7 was used with a *p*-value cutoff of 0.001[56]. For the coupling motifs, we supplied frequency matrices as previously published[50]. Overlapping motifs of the same type were merged, and overlapping motifs of different types were kept distinct. Results were plotted using the karyoploteR package in R (version 1.30.0)[57].

## Software

Confocal microscopy was done using Leica Application Suite X (LAS-X) version 4.6.1.27508. Image processing was done in ImageJ (version 1.54f with Java 1.8.0_322, 64-bit)[58,59], MorphoGraphX (version 2.0.1-394)[60], and R (version 4.3.1 (2023-06-16))[61] run in RStudio (Version 2023.12.0 + 369). Figures were assembled in Adobe Illustrator (version 28.6). An RGB color profile "Image P3" was used for all the figures.

## Statistical analyses

For bootstrapping estimates of distribution in Supplementary Fig. 7c, d, and f, from each sample, 10,000 bootstrap samples of the same size as the original sample were created. For two-sample permutation tests, two-sided *p*-values were calculated from 100,000 permutations. Two-sample, two-sided Wilcoxon's rank sum tests were done using wilcox.test() in R with default parameters. Linear regression was done using lm() in R. For multiple comparison, *p*-values from pairwise tests were Bonferroni-corrected and visualized using multcompLetters() in package "multcompView" (version 0.1-0)[62,63] in R.

## Reporting summary

Further information on research design is available in the Nature Portfolio Reporting Summary linked to this article.

# Data availability

Data supporting the findings of this work are available within the paper and its Supplementary Information files. A reporting summary for this Article is available as a Supplementary Information file. All plasmids and seeds are available upon request. Source data are provided with this paper.

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

## Acknowledgements

We thank Isabella Burda, Lanxi Hu, David Pan, Avilash Yadav, and Maura Zimmermann for comments on the manuscript. We thank Isabella Burda for the oryzalin treatment protocol. We thank Elliot Meyerowitz and Arnavaz Garda for the *DR5::3×VENUS-N7* seeds, and Dolf Weijers for the R2D2 (Col-Utr) seeds. We thank Frances Clark and Richard Smith for suggestions on image analyses. We thank VectorBuilder for making the constructs. Research reported in this publication was supported by the National Institute of General Medical Sciences of the National Institutes of Health (NIH) under award number R01GM134037 to A.H.K.R. and a Barbara McClintock Award from School of Integrative Plant Science, Cornell University to S.K. The content is solely the responsibility of the authors and does not represent the views of the National Institutes of Health.

## Author contributions

S.K.: Conceptualization, Investigation, Formal Analysis, Visualization, Writing - Original Draft, Writing - Review & Editing. B.R.: Investigation, Formal Analysis, Writing - Review & Editing. M.Z.: Conceptualization, Writing - Review & Editing. A.H.K.R: Conceptualization, Writing - Review & Editing, Supervision, Project administration, Funding acquisition.

## Competing interests

The authors declare no competing interests.
