## [Peer Review file · Nature Communications]

Stochastic Gene Expression in Auxin Signaling in the Floral Meristem of *Arabidopsis thaliana*

Corresponding Author: Dr Adrienne Roeder

Version 0:

Reviewer comments:

Reviewer #1

(Remarks to the Author)

The manuscript by Kong et al. entitled "Stochastic Gene Expression in Auxin Signaling in the Floral Meristem of *Arabidopsis thaliana*" reports enhanced stochastic expression of genes downstream of auxin signaling. The authors imaged promoter activity of DR5, promAHP6 and promDOF5.8 during sepal primordium initiation. They found higher variability of expression than ubiquitous promoters, such as 35S and promUBQ10. The variability becomes reduced following primordium development. Among them, synthetic DR5 has higher variability than endogenous promAHP6 and promDOF5.8. The authors used dual promoters to separate external and intrinsic noises, and proposed strong intrinsic noise, which is different from dominant external noise for ubiquitous 35S. Using DII/mDII, the authors showed that auxin level is likely more stable than DR5 signals. Together, this study expands our knowledge of stochastic gene expression, which has long been ignored by started to gain attentions in multicellular species. I have a few comments that I hope the authors found useful to improve the work.

1. The interpretation of oryzaline treatment as a means to increase cell size has obvious pitfalls. Oryzaline inhibits microtubule polymerization, and influences many cellular processes in addition to cell division. To manipulate cell division, there are other chemicals that specifically work on cell division steps. Alternatively, the authors can express cyclin and KIP-related proteins to promoter and inhibit cell proliferation, respectively.
2. Is the expression level positively correlated with variability? From Fig. 2c and 2d, as well as similar ones, it looks like not. More stringent statistics would be helpful to clarify the relationship.
3. What would happen after exogenous auxin treatment? Especially using auxin analogs insensitive to polar auxin transport (PAT), such as 2,4-D. This would exclude the contribution of variability originated from PAT. Note that PAT starts to drain auxin from sepal primordia during early development.

Reviewer #2

(Remarks to the Author)

The authors use a combination of imaging and (statistical) image analysis to reveal a previously under-appreciated aspect of developmental gene regulation in *Arabidopsis* flower buds - stochasticity. The work is superbly done (as is common in this team), and highly valuable to the community. While the work primarily addresses auxin response in flower buds, there is a potential that it has an impact on auxin-dependent gene regulation in a broader sense.

There are a few points to consider in improving the (very good) manuscript:

1. There is a resource that could have been used to further analyse the factors causing stochastic expression and variation in DR5 expression. In the Dr5v2 paper (Liao et al., Nature methods, 2015), the authors described a single-locus dual-DR5 transgene. Would this not be a good control for the effect of genomic insertion site on co-variation?
2. It is a bit unsatisfying that the degree by which AHP6 and DOF5.87 are influenced by noise is so much weaker than DR5. This truly makes DR5 an exception, which is not unexpected given its highly artificial architecture and lack of any additional promoter elements that could buffer expression or noise. I feel that this aspect, promoter architecture, is under-developed. At

the very least, it would be important to reflect on the topology of AuxRE's in the 3 promoters used, and discuss how their arrangement may correlate with the observed expression features. Also, some model for how ARF protein properties may relate to such behavior would be helpful.

Reviewer #3

(Remarks to the Author)

In "Stochastic Gene Expression in Auxin Signaling in the Floral Meristem of *Arabidopsis thaliana*", Kong, Zhu and Roeder proceed to demonstrate the presence, and investigate the role, of stochasticity in the expression of genes induced by auxin. To this aim, they rely on a series of imaging experiments using fluorescent reporters of auxin perception and auxin response. Overall the paper is written clearly and logically, making the results accessible and convincing.

In brief, the authors assess stochasticity of gene expression by means of a strategy previously used in other organisms, but never (it seems) in the context of auxin signalling. The approach consists in imaging tissues with two distinguishable reporters of the same activity, here two fluorescent reporters with separate wavelengths. By screening these signals at single-cell level on multiple replicates, correlated activity of the two reporters indicates noise driven by extrinsic factors, whereas lack of correlation indicates some stochasticity within single cells.

The auxin pathway involves a number of steps and the authors take advantage of reporters being available at two depth levels within that pathway. The R2D2 is indicative of the upstream step of auxin perception, as DII is degraded (almost) directly by auxin. On the other hand the widely-used DR5 reported accounts for transcriptional responses to auxin, at the downstream end of the pathway. As DR5 is a synthetic construct the authors rightfully consider two other responsee reporters which are naturally present in *Arabidopsis thaliana*: AHP6 and DOF5.8.

The perception of auxin, as reported by R2D2, does not display any significant stochasticity. On the other hand, downstream reporters show some clear intrinsic stochasticity; these are particularly noticeable with DR5, which on its own would raise questions about implications in planta. Although at lower levels, similar effects are seen for AHP6 and DOF5.8. The authors' observations show different stages, with a stochastic response in early stages which settles into more stable patterns later on. They are able to characterize these stages by refining stage 1 and 2 meristems into intermediary 1a-b and 2a-c stages defined in terms of quantifiable morphological features. Further quantifications show that the noise intensity at single-cell level does not change significantly over time, but becomes averaged out as the size (number of cells) becomes large enough on a contiguous zone of auxin response activity.

This study points at a newly observed phenomenon, without deciphering entirely its causes and effects. These would involve a lot of complexity and it seems fair to leave it as future research, which will no doubt benefit from the results and methods presented in this paper. This makes this paper a novel and useful contribution.

I could not identify any significant typos or shortcomings and would support publication in the current form.

Version 1:

Reviewer comments:

Reviewer #1

(Remarks to the Author)

The authors have adequately addressed all the concerns raised during the review process. I am pleased to endorse publication.

Reviewer #2

(Remarks to the Author)

The authors have addressed my limited concerns and I now enthusiastically support publication of this work.

Reviewer #3

(Remarks to the Author)

As mentioned in my previous review, I think this is a very interesting paper and support publication. This applies also to the current, enhanced version.

Reviewer #1 (Remarks to the Author):

The manuscript by Kong et al. entitled “Stochastic Gene Expression in Auxin Signaling in the Floral Meristem of *Arabidopsis thaliana*” reports enhanced stochastic expression of genes downstream of auxin signaling. The authors imaged promoter activity of DR5, promAHP6 and promDOF5.8 during sepal primodium initiation. They found higher variability of expression than ubiquitous promoters, such as 35S and promUBQ10. The variability becomes reduced following primordium development. Among them, synthetic DR5 has higher variability than endogenous promAHP6 and promDOF5.8. The authors used dual promoters to separate external and intrinsic noises, and proposed strong intrinsic noise, which is different from dominant external noise for ubiquitous 35S. Using DII/mDII, the authors showed that auxin level is likely more stable than DR5 signals. Together, this study expands our knowledge of stochastic gene expression, which has long been ignored by started to gain attentions in multicellular species. I have a few comments that I hope the authors found useful to improve the work.

Response: We thank Reviewer #1 for appreciation of our work and the helpful suggestions for improving it.

1. The interpretation of oryzaline treatment as a means to increase cell size has obvious pitfalls. Oryzaline inhibits microtubule polymerization, and influences many cellular processes in addition to cell division. To manipulate cell division, there are other chemicals that specifically work on cell division steps. Alternatively, the authors can express cyclin and KIP-related proteins to promoter and inhibit cell proliferation, respectively.

Response: Following your suggestion, we have tested roscovitine, olomoucine, and hydroxyurea. Of those, only hydroxyurea significantly decreased cell number in stage 2 meristems. Hydroxyurea-treated buds look more variable than mock, similar to oryzalin-treated buds, with sporadic auxin patches outside the incipient sepal regions. We observed a small (11%) increase in the variability of global pattern, while the cellular noise does not change. This increase is less significant than oryzalin (24%). We think this may be because hydroxyurea is less effective in decreasing cell number (33% decrease in 6 days vs. 59% decrease in 4 days). Notwithstanding, both experiments support the same qualitative conclusion that decreasing cell number of the patterning domain impairs the robustness of global pattern formation. The new results are added to a new figure (Supplementary Fig. 8).

2. Is the expression level positively correlated with variability? From Fig. 2c and 2d, as well as similar ones, it looks like not. More stringent statistics would be helpful to clarify the relationship.

Response: We used CV to quantify the variability, which is standard deviation divided by mean. This is why variability is not positively correlated with expression level in Fig. 2c and 2d. To help the readers, we have added this definition to the main text (Line 153-154).

3. What would happen after exogenous auxin treatment? Especially using auxin analogs insensitive to polar auxin transport (PAT), such as 2,4-D. This would exclude the contribution of variability originated from PAT. Note that PAT starts to drain auxin from sepal primordia during early development.

Response: Following your suggestion, we treated meristems with 2,4-D. Such treatment does not abolish the variability in DR5 expression pattern. In stage 1, there are still sporadic patches of high DR5 expression. In stage 2, in contrast to the expectation that all cells of the peripheral zone would show uniformly high DR5 expression, DR5 is in fact still highly variable. Quantification shows that CV decreases by ~19.7%-37.8% in early-stage meristems, which may reflect either a minor contribution of variability originated from PAT, or merely reflecting a higher mean expression of DR5 in 2,4-D (because CV is standard deviation divided by mean). These results are added to a new figure (Supplementary Fig. 2).

In either case, if PAT is variable, such variability is likely averaged out at the step of auxin perception by TIR1/AFB, which we show is mostly robust (Fig. 3). Overall, our experiments suggest that processes upstream to auxin-responsive gene expression, such as PAT, auxin level fluctuations, and heterogeneity in auxin perception, only makes a minor contribution at most to the variability in DR5 patterns.

Reviewer #2 (Remarks to the Author):

The authors use a combination of imaging and (statistical) image analysis to reveal a previously under-appreciated aspect of developmental gene regulation in Arabidopsis flower buds - stochasticity. The work is superbly done (as is common in this team), and highly valuable to the community. While the work primarily addresses auxin response in flower buds, there is a potential that it has an impact on auxin-dependent gene regulation in a broader sense.

Response: We thank Reviewer #2 for the nice comments and suggestions for improving our manuscript.

There are a few points to consider in improving the (very good) manuscript:

1. There is a resource that could have been used to further analyse the factors causing

stochastic expression and variation in DR5 expression. In the Dr5v2 paper (Liao et al., Nature methods, 2015), the authors described a single-locus dual-DR5 transgene. Would this not be a good control for the effect of genomic insertion site on co-variation?

Response: Thank you for pointing out this resource. We did not use it because the double reporter in Liao et al. used two different DR5 promoters (DR5v2 and DR5rev). Thus, although the two reporters are in the same locus, expression differences between them should reflect promoter differences in addition to stochastic gene expression.

2. It is a bit unsatisfying that the degree by which AHP6 and DOF5.7 are influenced by noise is so much weaker than DR5. This truly makes DR5 an exception, which is not unexpected given its highly artificial architecture and lack of any additional promoter elements that could buffer expression or noise. I feel that this aspect, promoter architecture, is under-developed. At the very least, it would be important to reflect on the topology of AuxRE's in the 3 promoters used, and discuss how their arrangement may correlate with the observed expression features. Also, some model for how ARF protein properties may relate to such behavior would be helpful.

Response: Indeed, the discussion related to promoter elements and architecture could have been better developed. In the revised manuscript, we included a new figure (Supplementary Fig. 9) which shows the arrangement of AuxREs and coupling elements in the promoters of DR5, AHP6, and DOF5.8. We have extended our discussion on this topic, as follows (Line 426-440).

“*pDR5* consists of nine tandem TGTCTC repeats in reverse orientation sandwiching eight pyrimidine-rich motifs (Y-patches) proximal (-191 to -98 bp) to the transcription start site (TSS). Such close packing of AuxREs may confer high affinity to ARFs, facilitating their dimerization and permitting sporadic transcription. On the other hand, *pAHP6* and *pDOF5.8* contain fewer AuxREs, which are more widely spaced and all outside the proximal promoter (-250 to +1 bp). Such arrangement may set a higher threshold for ARF-mediated transcription and serve to filter out stochastic noise. Moreover, *pAHP6* and *pDOF5.8* contains diverse AuxREs, including TGTCGG, TGTCAC, and TGTSTBC, in addition to TGTCTC which is the only AuxRE in *pDR5*. Such diversity may promote the heterodimerization between ARFs with different AuxRE preferences, thus filtering out noise from the stochastic fluctuation of the level of a particular ARF. Lastly, *pAHP6* and *pDOF5.8* have a wide range of coupling elements surrounding the AuxREs. These coupling elements can serve as binding sites for ARF partners, which may exert control in noise level. Future studies are needed to better understand the effect of promoter architecture on the stochasticity of gene expression during multicellular development.”

Reviewer #3 (Remarks to the Author):

In "Stochastic Gene Expression in Auxin Signaling in the Floral Meristem of *Arabidopsis thaliana*", Kong, Zhu and Roeder proceed to demonstrate the presence, and investigate the role, of stochasticity in the expression of genes induced by auxin. To this aim, they

rely on a series of imaging experiments using fluorescent reporters of auxin perception and auxin response. Overall the paper is written clearly and logically, making the results accessible and convincing.

In brief, the authors assess stochasticity of gene expression by means of a strategy previously used in other organisms, but never (it seems) in the context of auxin signalling. The approach consists in imaging tissues with two distinguishable reporters of the same activity, here two fluorescent reporters with separate wavelengths. By screening these signals at single-cell level on multiple replicates, correlated activity of the two reporters indicates noise driven by extrinsic factors, whereas lack of correlation indicates some stochasticity within single cells.

The auxin pathway involves a number of steps and the authors take advantage of reporters being available at two depth levels within that pathway. The R2D2 is indicative of the upstream step of auxin perception, as DII is degraded (almost) directly by auxin. On the other hand the widely-used DR5 reporter accounts for transcriptional responses to auxin, at the downstream end of the pathway. As DR5 is a synthetic construct the authors rightfully consider two other responsee reporters which are naturally present in *Arabidopsis thaliana*: AHP6 and DOF5.8.

The perception of auxin, as reported by R2D2, does not display any significant stochasticity. On the other hand, downstream reporters show some clear intrinsic stochasticity; these are particularly noticeable with DR5, which on its own would raise questions about implications in planta. Although at lower levels, similar effects are seen for AHP6 and DOF5.8.

The authors' observations show different stages, with a stochastic response in early stages which settles into more stable patterns later on. They are able to characterize these stages by refining stage 1 and 2 meristems into intermediary 1a-b and 2a-c stages defined in terms of quantifiable morphological features. Further quantifications show that the noise intensity at single-cell level does not change significantly over time, but becomes averaged out as the size (number of cells) becomes large enough on a contiguous zone of auxin response activity.

This study points at a newly observed phenomenon, without deciphering entirely its causes and effects. These would involve a lot of complexity and it seems fair to leave it as future research, which will no doubt benefit from the results and methods presented in this paper. This makes this paper a novel and useful contribution.

I could not identify any significant typos or shortcomings and would support publication in the current form.

Response: We appreciate your precise summary of our work and its potential contributions to the field. Indeed, we expect our work to open up new avenues for studying the precise cause of the observed stochasticity in auxin signaling and its effects on plant development.